# How much is a noisy image worth?
# Data Scaling Laws for Ambient Diffusion.

**Giannis Daras**[*]
CSAIL
MIT, IFML
gdaras@mit.edu

**Yeshwanth Cherapanamjeri**[*]
CSAIL
MIT
yesh@mit.edu

**Costantinos Daskalakis**
CSAIL
MIT
costis@csail.mit.edu

## Abstract

The quality of generative models depends on the quality of the data they are trained on. Creating large-scale, high-quality datasets is often expensive and sometimes impossible, e.g. in certain scientific applications where there is no access to clean data due to physical or instrumentation constraints. Ambient Diffusion and related frameworks train diffusion models with solely corrupted data (which are usually cheaper to acquire) but ambient models significantly underperform models trained on clean data. We study this phenomenon at scale by training more than $80$ models on data with different corruption levels across three datasets ranging from $30,000$ to $\approx 1.3M$ samples. We show that it is impossible, at these sample sizes, to match the performance of models trained on clean data when only training on noisy data. Yet, a combination of a small set of clean data (e.g. $10\%$ of the total dataset) and a large set of highly noisy data suffices to reach the performance of models trained solely on similar-size datasets of clean data, and in particular to achieve near state-of-the-art performance. We provide theoretical evidence for our findings by developing novel sample complexity bounds for learning from Gaussian Mixtures with heterogeneous variances. Our theoretical model suggests that, for large enough datasets, the effective marginal utility of a noisy sample is exponentially worse that of a clean sample. Providing a small set of clean samples can significantly reduce the sample size requirements for noisy data, as we also observe in our experiments.

## 1 Introduction

A key factor behind the remarkable success of modern generative models, from image Diffusion Models (DMs) to Large Language Models (LLMs), is the curation of large scale datasets (Gadre et al., 2023; Li et al., 2024). However, in certain applications, access to high-quality data is scarce, expensive, or impossible. For example, in Magnetic Resonance Imaging (MRI) the quality of the data is proportional to the time spent in the scanner (Jalal et al., 2021) and, in black-hole imaging, it is never possible to get full measurements from the object of interest (Lin et al., 2024). Constructing a (copyright-free) large-scale dataset of high-quality general domain images is also an expensive and complex process. Enterprise text-to-image image DMs rely on proprietary datasets, often acquired from third-party vendors at significant cost (Betker et al., 2023; Imagen-Team-Google et al., 2024). State-of-the-art open-source DMs are typically trained by crawling a large pool of images from the Web and filtering them for quality with a pipeline that deems each sample as suitable or non-suitable for training (Gadre et al., 2023). However, this binary treatment of samples is problematic because often low-quality images still contain useful information. For example, a blurry image might get dismissed from the filtering pipeline to avoid blurry generations at inference time, yet the image might still contain important information about the world, such as the type of objects present at the scene.

Recently, there has been a growing interest in developing frameworks for training generative models using corrupted data, e.g. from blurry or noisy images (Bora et al., 2018; Daras et al., 2023b; Kawar et al., 2023; Aali et al., 2023; Daras et al., 2024; Bai et al., 2024; Rozet et al., 2024; Wang et al., 2024; Kelkar et al., 2023). Most frameworks that are applicable to diffusion models introduce training or sampling approximations, which significantly hurt performance. A recent work, *Consistent Diffusion Meets Tweedie* (Daras et al., 2024), proposes a method that guarantees exact sampling, at least asymptotically as the number of noisy samples tends to infinity. For realistic sample sizes, however,

---

[*]Equal contribution.

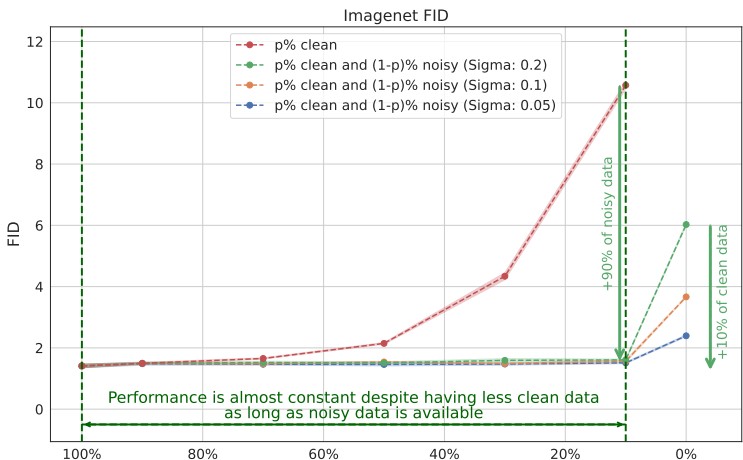

Figure 1: ImageNet FID performance (lower is better) for models trained with different amounts of clean and noisy data. Performance of models trained with only clean data (red curve) reduces as we decrease the amount of data used. Training with purely noisy data (rightmost points) also gives poor performance – even if 100% of the dataset is available. Training with a mix of noisy and clean data strikes an interesting balance: *a model trained with* 90% *noisy and* 10% *clean data is almost as good as a model trained with* 100% *clean data.*

models trained on noisy data using this method (or any other proposed method) are significantly inferior to models trained on clean data. How do we overcome this challenge?

A fundamental limitation facing existing approaches for leveraging corrupted data is that, so far, they have been studied under the assumption that no access to clean samples is available. As we show in this work, it is theoretically and experimentally impossible to compensate for the lack of clean data without a significant hit (that is *exponential in the number of classes* in our theoretical analysis) in the number of noisy datapoints. On the other hand, we show that a combination of a small set of clean images (e.g. 10% of the total dataset) and a large set of noisy images (e.g. 90% of the total dataset) suffices to maintain state-of-the-art performance. Models trained on this data mixture significantly outperform two natural baselines: i) training only on the small set of clean images; and ii) training only on the large set of noisy images (see also Figure 1 and Table 1). We validate our finding by training more than 80 models across various noise scales and corruption ratios on ImageNet, CelebA-HQ, and CIFAR-10.

Theoretically, we consider a novel variant of the classical problem of learning Gaussian Mixture Models (GMMs) with *heterogeneous* noise levels. In contrast to the classical setting, each sample is generated with a (potentially) different noise level reflective of how clean or noisy the data point is. We derive minimax optimal estimation rates, which showcase the differing utility of clean and noisy data. Concretely, having lots of noisy data enables identification of the right low-dimensional structure for learning the

| Available Data | CIFAR-10 | CelebA | ImageNet |
|---|---|---|---|
| 100% clean | $1.99_{\pm 0.02}$ | $2.40_{\pm 0.07}$ | $1.41_{\pm 0.03}$ |
| 100% noisy | $11.93_{\pm 0.09}$ | $12.97_{\pm 0.11}$ | $5.32_{\pm 0.08}$ |
| 10% clean (smaller dataset) | $17.30_{\pm 0.14}$ | $11.92_{\pm 0.05}$ | $10.57_{\pm 0.06}$ |
| 90% noisy + 10% clean | $2.81_{\pm 0.02}$ | $2.75_{\pm 0.02}$ | $1.68_{\pm 0.02}$ |

Table 1: FID scores for models trained on datasets with different percentages of noisy and clean data. Training with a mix of 90% noisy and 10% clean data achieves much better performance than training the same number of purely noisy samples, or just the clean samples.

mixture components, while a more moderate number of clean data allows fine-grained differentiation of the components in low dimensions. In particular, effective learning is possible even when the number of clean samples is substantially *smaller* than the dimension. At the same time, relying solely on noisy data *requires* exponentially (in the number of components) noisy samples than clean ones. This theoretical result could justify why training on a mixture of clean and noisy data significantly outperforms training solely on either one of the clean or the noisy sets. In summary, **our contributions** are as follows:

- We train more than 80 models at different datasets, noise levels and corruption ratios.
- We show that as long as a few clean images are available, models trained with corrupted data perform on par with models trained on clean data. We further show that training on a combination of a small set of clean images and a large set of noisy images significantly outperforms training solely on either one of these sets.
- We provide a theoretical explanation for this phenomenon by developing novel sample complexity bounds for the case of GMMs with heterogeneous variances. Our minimax rates show that the availability of noisy data removes the *polynomial* dependence on dimension in the number of clean samples while simultaneously showing *exponentially* more noisy samples are required to compensate for the lack of clean data.

- Through our theoretical analysis and our experimental validation, we pinpoint the price of noisy images for different datasets and noise levels. These insights can potentially guide budget allocation for dataset curation.

## 2 BACKGROUND AND RELATED WORK

### 2.1 DIFFUSION MODELING

The dominant paradigm for learning distributions is diffusion models. A diffusion model is trained to denoise images at different levels of additive Gaussian noise (Ho et al., 2020; Song & Ermon, 2019; Song et al., 2020b). To establish some notation, we will use $t \in [0, T]$ for the diffusion time and $\sigma_t$ for the associated standard deviation of the added noise. Let $\boldsymbol{X}_0 \sim p_0$ be a sample from the distribution of interest and let $\boldsymbol{X}_t \sim p_t$ a sample of the noisy distribution at time $t$, where it holds that $\boldsymbol{X}_t = \boldsymbol{X}_0 + \sigma_t \boldsymbol{Z}, \boldsymbol{Z} \sim \mathcal{N}(\boldsymbol{0}, I)$. At training time, the goal is to learn $\mathbb{E}[\boldsymbol{X}_0 | \boldsymbol{X}_t = \boldsymbol{x}_t]$ for all levels $t$. Diffusion models are trained to approximate this quantity using the following objective[1] (Vincent, 2011):

$$J_{\mathrm{DSM}}(\boldsymbol{\theta}) = \mathbb{E}_{\boldsymbol{x}_0 \sim p_0} \mathbb{E}_{t \sim U[0,T]} \mathbb{E}_{\boldsymbol{x}_t \sim p_t(\boldsymbol{x}_t | \boldsymbol{X}_0 = \boldsymbol{x}_0)} \| \boldsymbol{h}_{\boldsymbol{\theta}}(\boldsymbol{x}_t, t) - \boldsymbol{x}_0 \|^2 . \tag{1}$$

Once trained, the model is used for sampling following a discretized version of the update rule (Song et al., 2020b;a):

$$\mathrm{d}\boldsymbol{x}_t = -\mathrm{d}\sigma_t \frac{\boldsymbol{h}_{\boldsymbol{\theta}}(\boldsymbol{x}_t, t) - \boldsymbol{x}_t}{\sigma_t}, \tag{2}$$

that is guaranteed to sample from $p_0$ given that it is initialized from $p_T$ and $\boldsymbol{h}_{\boldsymbol{\theta}}(\boldsymbol{x}_t, t) = \mathbb{E}[\boldsymbol{X}_0 | \boldsymbol{X}_t = \boldsymbol{x}_t]$. In practice, there are errors due to initialization, discretization, and improper learning of the score (Chen et al., 2022; 2023).

### 2.2 LEARNING DIFFUSION MODELS FROM NOISY DATA

The training objective of Equation 1 requires access to data from the distribution $p_0$. We will refer to this data as *clean* for the rest of this paper. As explained in the Introduction, in many applications access to clean data is scarce and hence methods for training diffusion models from corrupted data have been developed.

Ambient Diffusion (Daras et al., 2023b) proposes a framework to train diffusion models from linear measurements. An alternative framework to Ambient Diffusion leverages Stein's Unbiased Risk Estimate (SURE) (Kawar et al., 2023; Aali et al., 2023). These works extend classical results for training *restoration* models from corrupted data (Moran et al., 2020; Stein, 1981; Xu et al., 2020; Pang et al., 2021; Krull et al., 2019; Batson & Royer, 2019) to the setting of learning *generative* models under corruption. Unfortunately, these methods induce sampling approximations that significantly hurt the performance. A different thread of papers on this topic relies on the Expectation-Maximization (EM) algorithm (Bai et al., 2024; Rozet et al., 2024; Wang et al., 2024). The convergence of these models to the true distribution depends on the convergence of the EM algorithm, which might get stuck in a local minimum.

Consistent Diffusion Meets Tweedie (Daras et al., 2023a; 2024) generalized Ambient Diffusion to handle additive Gaussian noise and became the first framework with provably exact sampling given infinite (noisy) data. In what follows, we give a brief description of the method, since we are going to use this in the rest of this paper.

**Problem Setting.** Assume that we are given samples at a specific noise level $t_n$ (nature noise level). The question becomes how we can leverage these samples to learn the optimal denoiser $\mathbb{E}[\boldsymbol{X}_0 | \boldsymbol{X}_t = \boldsymbol{x}_t]$ for all noise levels $t$.

**Learning the optimal denoiser for high noise levels.** The first task is to learn the optimal denoiser for $t > t_n$. To perform this goal, one can leverage the following Lemma:

**Lemma 2.1** ((Daras et al., 2024)). *Let* $\boldsymbol{X}_{t_n} = \boldsymbol{X}_0 + \sigma_{t_n} \boldsymbol{Z}_1$ *and* $\boldsymbol{X}_t = \boldsymbol{X}_0 + \sigma_t \boldsymbol{Z}_2, \quad \boldsymbol{Z}_1, \boldsymbol{Z}_2 \sim \mathcal{N}(\boldsymbol{0}, I)$ *i.i.d. Then, for any* $\sigma_t > \sigma_{t_n}$, *we have that:*

$$\underbrace{\mathbb{E}[\boldsymbol{X}_0 | \boldsymbol{X}_t = \boldsymbol{x}_t]}_{\text{removes all the noise}} = \frac{\sigma_t^2}{\sigma_t^2 - \sigma_{t_n}^2} \underbrace{\mathbb{E}[\boldsymbol{X}_{t_n} | \boldsymbol{X}_t = \boldsymbol{x}_t]}_{\text{removes additional noise}} - \frac{\sigma_{t_n}^2}{\sigma_t^2 - \sigma_{t_n}^2} \boldsymbol{x}_t. \tag{3}$$

---

[1]In practice, a non-uniform distribution for $t$ and a loss weighting $w(t)$ are often used (Karras et al., 2022).

This Lemma states that to remove all the noise there is, it suffices to remove the noise that we added on top of the dataset noisy samples from the R.V. $\boldsymbol{X}_{t_n}$. One can train the denoiser $\mathbb{E}[\boldsymbol{X}_{t_n}|\boldsymbol{X}_t = \boldsymbol{x}_t]$ with supervised learning, as in Equation 1. The following loss will train the network to estimate $\mathbb{E}[\boldsymbol{X}_0|\boldsymbol{X}_t = \boldsymbol{x}_t]$ directly:

$$J_{\text{Ambient DSM}}(\theta) = \mathbb{E}_{\boldsymbol{x}_{t_n} \sim p_{t_n}} \mathbb{E}_{t \sim U(t_n,T]} \mathbb{E}_{\boldsymbol{x}_t \sim p_t(\boldsymbol{x}_t|\boldsymbol{X}_{t_n}=\boldsymbol{x}_{t_n})} \left\| \frac{\sigma_t^2 - \sigma_{t_n}^2}{\sigma_t^2} \boldsymbol{h}_\theta(\boldsymbol{x}_t, t) + \frac{\sigma_{t_n}^2}{\sigma_t^2} \boldsymbol{x}_t - \boldsymbol{x}_{t_n} \right\|^2. \quad (4)$$

This idea is closely related to Noisier2Noise (Moran et al., 2020). We underline that there are also alternative ways to learn the optimal denoiser in this regime, such as SURE (Stein, 1981). However, SURE-based methods usually bring a computational overhead, as one needs to compute (or approximate) a Jacobian Vector Product.

**Learning the optimal denoiser for low noise levels.** Learning the optimal denoiser for $t < t_n$ is a significantly harder problem since there is no available data in this regime. Most of the prior works did not even attempt to solve this problem. At training time, a denoiser $\boldsymbol{h}_\theta(\boldsymbol{x}_t, t)$ is trained for all $t \geqslant t_n$. At sampling time, one can: i) also use $\boldsymbol{h}_\theta$ for times $t < t_n$, even though the model was not trained in this regime, hoping that the neural network will extrapolate in a meaningful way, or ii) truncate the sampling at time $t_n$, by predicting $\hat{\boldsymbol{x}}_0 = \boldsymbol{h}_\theta(\boldsymbol{x}_{t_n}, t_n)$. In Algorithm 2 of the Appendix, we provide a reference implementation of the truncated sampling idea. A more principled attempt was introduced by Daras et al. (2024) where the authors attempt to learn the optimal denoiser for low noise levels by enforcing a consistency loss. Specifically, for $t < t_n$, the network is trained with the following objective:

$$J_{\text{C}}(\theta) = \mathbb{E}_{t,t',t'' \sim \mathcal{U}(t_n,T],\mathcal{U}(\epsilon,t),\mathcal{U}(t'-\epsilon,t')} \mathbb{E}_{\boldsymbol{x}_t} \mathbb{E}_{\boldsymbol{x}_{t'}|\boldsymbol{x}_t} \left[ \left\| \boldsymbol{h}_\theta(\boldsymbol{x}_{t'}, t') - \mathbb{E}_{\boldsymbol{x}_{t''} \sim p_\theta(\boldsymbol{x}_{t''}, t''|\boldsymbol{x}_{t'}, t')} \left[ \boldsymbol{h}_\theta(\boldsymbol{x}_{t''}, t'') \right] \right\|^2 \right], \quad (5)$$

where $p_\theta$ is the distribution induced by the SDE version of Equation 2 (we refer the interested reader to the works of Song et al. (2020b), Daras et al. (2023a)). Assuming that the network has been perfectly trained to predict $\mathbb{E}[\boldsymbol{X}_0|\boldsymbol{X}_t = \boldsymbol{x}_t]$ for all times $t \geqslant t_n$, the minimizer of this loss will be $\mathbb{E}[\boldsymbol{X}_0|\boldsymbol{X}_t = \boldsymbol{x}_t]$ also for times $t \leqslant t_n$. There are a few important drawbacks though. First, the consistency loss adds a significant training overhead since at each training step: i) expectations need to be computed and ii) sampling steps need to be performed. Second, the theoretical guarantees only hold given perfect learning of the conditional expectation for $t \geqslant t_n$ and it is unclear how errors propagate. Finally, even for slight corruption, the method leads to a significant performance deterioration (Daras et al., 2024).

## 3 METHOD

**Problem Setting.** The need for consistency or sampling approximations arises from the fact that we do not have samples to train the network in the regime $t \in (t_n, 0]$. We relax this assumption by considering the setting where a small set of clean and a large set of noisy images are available. To establish the notation, let's use $S_{\text{clean}}$ to denote the clean set and $S_{\text{noisy}}$ to denote the noisy set. The clean samples are realizations of the R.V. $\boldsymbol{X}_0 \sim p_0$ and the noisy samples realizations of the R.V. $\boldsymbol{X}_{t_n} = \boldsymbol{X}_0 + \sigma_{t_n} \boldsymbol{Z}$ for some fixed noise level $t_n$.

**Training algorithm.** Our training algorithm leverages samples from both sets. For times $t \in [t_n, 0]$, we will rely solely on clean samples. On the other hand, for $t \in [T, t_n)$, we will leverage *both* the clean and the noisy samples. The clean samples will be used as usual, following the Denoising Score Matching objective, defined in Equation 1. The noisy samples in this regime will utilize Equation 4 to learn a denoiser by relating it to the quantity $\mathbb{E}[\boldsymbol{X}_{t_n}|\boldsymbol{X}_t = \boldsymbol{x}_t]$ through the use of Lemma 2.1 (see Algorithm 1).

**Method Discussion.** In prior work (Daras et al., 2024), the optimal denoiser for times $t < t_n$ was learned through the (expensive) consistency loss. In our setting, we assume access to a small set of clean images that we are using to learn the optimal denoiser for these times. The question then becomes: "*does a small set of clean images suffice to learn the optimal denoiser for low-noise levels?*" As we show in this paper, the answer to this question is affirmative, i.e. we only need a handful of uncorrupted images to learn the optimal denoiser for low corruption.

The intuition is as follows. To learn the distribution, we need to be able to generate accurately both structural information (low-frequencies) and texture (high-frequencies). Clean samples have both low-frequency and high-frequency content. The addition of noise corrupts the high-frequency content faster (for a discussion, see Dieleman (2024)). Hence, noisy images are mostly useful to learn the low-frequency content of the distribution, i.e. they are useful to learn to generate structure. Humans, and metrics that are trying to model human preferences (such as FID), are more

---

**Algorithm 1** Algorithm for training using both clean and noisy samples.

---

**Input:** untrained network $\boldsymbol{h}_\theta$, set of clean samples $S_{\text{clean}}$, set of noisy samples $S_{\text{noisy}}$, std of the noise $\sigma_{t_n}$, batch size
    $B$, diffusion time $T$, noisy diffusion start time $t_n$

1: **while** not converged **do**
2:     Form a batch $\mathcal{B}$ of size $B$ uniformly sampled from $S_{\text{clean}} \cup S_{\text{noisy}}$
3:     loss $\leftarrow 0$                                                               ▷ Initialize loss.
4:     **for** each sample $x \in \mathcal{B}$ **do**
5:         $\epsilon \sim \mathcal{N}(\mathbf{0}, I)$                                        ▷ Sample noise.
6:         **if** $x \in S_{\text{noisy}}$ **then**
7:             $\boldsymbol{x}_{t_n} \leftarrow \boldsymbol{x}$                 ▷ We are dealing with a noisy sample.
8:             $t \sim \mathcal{U}(t_n, T)$         ▷ Sample diffusion time for noisy sample.
9:             $\boldsymbol{x}_t = \boldsymbol{x}_{t_n} + \sqrt{\sigma_t^2 - \sigma_{t_n}^2}\,\epsilon$              ▷ Add additional noise.
10:             loss $\leftarrow$ loss $+ \left\| \frac{\sigma_t^2 - \sigma_{t_n}^2}{\sigma_t^2} \boldsymbol{h}_\theta(\boldsymbol{x}_t, t) + \frac{\sigma_{t_n}^2}{\sigma_t^2} \boldsymbol{x}_t - \boldsymbol{x}_{t_n} \right\|^2$   ▷ Ambient Denoising Score Matching loss.
11:         **else**
12:             $\boldsymbol{x}_0 \leftarrow \boldsymbol{x}$                    ▷ We are dealing with a clean sample.
13:             $t \sim \mathcal{U}(0, T)$          ▷ Sample diffusion time for clean sample.
14:             $\boldsymbol{x}_t = \boldsymbol{x}_0 + \sigma_t \epsilon$                   ▷ Add additional noise.
15:             loss $\leftarrow$ loss $+ \|\boldsymbol{h}_\theta(\boldsymbol{x}_t, t) - \boldsymbol{x}_0\|^2$     ▷ Regular Denoising Score Matching loss.
16:         **end if**
17:     **end for**
18:     loss $\leftarrow \frac{\text{loss}}{B}$                                         ▷ Compute mean loss.
19:     $\theta \leftarrow \theta - \eta \nabla_\theta \text{loss}$       ▷ Update network parameters via backpropagation.
20: **end while**

---

perceptible to errors in structural information than errors in the high-entropy image texture. So even though it might be impossible to perfectly generate high-frequency content using only a few clean images, metrics, such as FID, are going to penalize these errors less, given that the model can accurately generate the low-frequency content.

The method we developed in Algorithm 1 can be trivially extended to more than two sets of points. In fact, each point $\boldsymbol{x}_i$ in the training set can have its own noise level $t_i$. In that case, our algorithm would use each point $\boldsymbol{x}_i$ to learn the optimal denoiser for $t \geqslant t_i$. For the ease of presentation, we stick to two noise levels for the rest of this paper. Further, if the noise level $\sigma_{t_n}$ is not known, it can be estimated by computing the difference of the variances between the sets $S_{\text{clean}}$ and $S_{\text{noisy}}$.

## 4   Theoretical Results

Before presenting our experiments, we establish theoretical results that will support our experimental findings. We note that most theoretical works on distribution learning operate under the assumption of i.i.d samples from a fixed distribution. In contrast, the goal of our paper is to understand the effects that heterogeneity in sample *quality* has on estimation error. This setting is quite niche and creates space for interesting theory to be developed.

We develop our results for the Gaussian Mixture Model that is commonly used in diffusion theory (Shah et al., 2023; Gatmiry et al., 2024; Gu et al., 2024; Chidambaram et al., 2024; Li & Chen, 2024; Cui et al., 2023; Chen et al., 2024). We examine the following adaptation of the classical estimation problem to the heterogeneous setting:

**Definition 4.1.** We are given $X_1, \dots, X_n$ independently drawn from the following distribution:

$$X_i \sim \mathcal{D} * \mathcal{N}(0, \sigma_i^2 I), \text{ where } \mathcal{D} = \sum_{i=1}^{k} w_i \delta_{\mu_i}$$

where $\delta_\mu$ denotes the Dirac distribution at $\mu$ and $*$ the convolution operator. The goal is to output $\widehat{\mathcal{D}}$ minimizing:

$$\mathrm{W}_1(\mathcal{D}, \widehat{\mathcal{D}}) \coloneqq \min_{C(\mathcal{D}, \widehat{\mathcal{D}})} \mathbb{E}_{(X, X') \sim C}[\|X - X'\|]$$

where $C(\mathcal{D}, \widehat{\mathcal{D}})$ is a coupling of $\mathcal{D}, \widehat{\mathcal{D}}$ with marginals equal to $\mathcal{D}$ and $\widehat{\mathcal{D}}$.

Note now that variance in sample quality can now be naturally expressed in terms of the noise level, $\sigma_i$, of the individual observations with the larger $\sigma_i$ denoting lower-quality samples and vice-versa.

## 4.1 MAIN THEORETICAL RESULTS AND DISCUSSION

**Overview of Theoretical Results.** Theorems 4.2 and 4.4 and Corollary 4.3 establish tight bounds on the sample complexity of learning a heterogeneous mixture-of-Gaussians model which precisely characterize the utilities of high and low-quality samples. These theoretical results match the intuition from Section 3 where noisy data is useful for learning low-frequency information while clean data helps in resolving high-frequency details. Concretely, the low-dimensional subspace spanned by the means of the GMM model corresponds to the low-frequency information estimated from the noisy samples while the resolution of the mixture components in the lower-dimensional space corresponds to the high-frequency information extracted from the clean samples. Hence, the heterogeneous mixture of Gaussians model provides a concrete instantiations of the intuition previously presented. These results further define the notion of the effective number of samples which captures how much actual information is in our dataset accounting for the per-sample corruption. This notion motivates the data pricing questions asked in Subsection 5.3.

*# samples*   *dimension*   *# classes*

**Theorem 4.2** (Upper bound). *Let $n$, $d$, $k$ $\in \mathbb{N}$, $R > 0$, and $\delta \in (0, 1/2)$. Suppose $\mathcal{D} = \sum_{i=1}^{k} w_i \delta_{\mu_i}$ is a $k$-atomic distribution over $\mathbb{R}^d$ with $\|\mu_i\| \leqslant R$. Then, there is a procedure (Algorithm 8) which when given $n$ independent samples $X_1, \ldots, X_n$ such that $X_i \sim \mathcal{D} * \mathcal{N}(0, \sigma_i^2 I)$ with $\sigma_i \geqslant 1$ satisfying:*

*# effective samples*

$$\max_i \frac{1}{\sigma_i^4} \leqslant \frac{c_1}{(d + \log(1/\delta))} \sum_{i=1}^{n} \frac{1}{\sigma_i^4}, \quad \max_i \frac{1}{\sigma_i^{4k-2}} \leqslant \frac{c_2}{\log(1/\delta) + k\log(k)} \sum_{i=1}^{n} \frac{1}{\sigma_i^{4k-2}}$$

*and $\delta > 0$, returns an estimate $\widehat{\mathcal{D}}$ with:*

*dimensionality reduction error*   *low-dim estimation error*

$$\mathrm{W}_1(\mathcal{D}, \widehat{\mathcal{D}}) \leqslant C \left( k \left( \frac{d + \log(1/\delta)}{\sum_{i=1}^{n} 1/\sigma_i^4} \right)^{1/4} + k^3 \left( \frac{k \log k + \log(1/\delta)}{\sum_{i=1}^{n} 1/\sigma_i^{4k-2}} \right)^{1/(4k-2)} \right),$$

*with probability at least $1 - \delta$. Furthermore, the runtime of the algorithm is $\widetilde{O}_k(nd)$.*

**Discussion.** The full proof of this result is given in Appendix C. We adopt a two-stage procedure where in the first stage we reduce the dimensionality of the problem, from the dimension $d$, to the number of components, $k$, by estimating the sub-space containing the means, $\mu_i$, and projecting all our data points onto that subspace. Next, we run a low-dimensional estimation procedure on the $k$-dimensional projected data (Algorithm 7) to estimate the means within the subspace. The two terms in the error bound reflect the contributions of each of these steps. The first corresponds to the error in estimating the low-dimensional sub-space and the second corresponds to the error of the low-dimensional estimation procedure. The two conditions (on $\max 1/\sigma_i^4$ and $\max 1/\sigma_i^{4k-2}$) lower bound the *effective* number of samples for the dimensionality-reduction and low-dimensional estimation steps respectively. The first requires at least $d$ (effective) samples for dimensionality reduction while at least $k$ are available for low-dimensional estimation.

Observing the two terms of the bound, we see that, compared to the high-quality points, the low-quality points have relatively more utility in the dimensionality-reduction step versus the lower-dimensional estimation step. This observation is made clearer in the following corollary in the simplified setting with 2 noise levels.

*Percentage of "clean" points*

**Corollary 4.3.** *Assume the setting of Theorem 4.2. Furthermore, suppose that for $p \in [0, 1]$, we additionally have that $\sigma_1, \ldots, \sigma_{pn} = 1$ and $\sigma_{pn+1}, \ldots, \sigma_n = \sigma \geqslant 1$. Then, the guarantees of Theorem 4.2 reduce to:*

$$\mathrm{W}_1(\mathcal{D}, \widehat{\mathcal{D}}) \leqslant C \left( k \left( \frac{d + \log(1/\delta)}{n(p + (1-p)/\sigma^4)} \right)^{1/4} + k^3 \left( \frac{k \log k + \log(1/\delta)}{n \left( p + (1-p)/\sigma^{4k-2} \right)} \right)^{1/(4k-2)} \right).$$

*Factor for dim. reduction*   *Factor for fine-grained estimation*

We see that the noisy samples are downweighted only by $1/\sigma^4$ for the dimensionality-reduction step whereas for the fine-grained estimation procedure, its discounting is much more substantial at $1/\sigma^{4k-2}$. Next, we present a matching lower bound establishing that the rate achieved by Theorem 4.2 is optimal for learning with heterogenous samples.

**Theorem 4.4** (Lower bound). *Let $k \in \mathbb{N}$. Let $n, k, d \in \mathbb{N}$, $\delta \in (0, 1/4]$, and $\{\sigma_i\}_{i=1}^n \geqslant 1$ be such that $d \geqslant 3$ and:*

$$\sum_{i=1}^n \frac{1}{\sigma_i^{4k-2}} \geqslant (Ck)^{2k-1} \log(1/\delta) \text{ and } \sum_{i=1}^n \frac{1}{\sigma_i^4} \geqslant C(d + \log(1/\delta))$$

*For any estimator $\widehat{T}$, there exists a $k$-atomic distribution $\mathcal{D}$ supported on $\mathbb{B}_1$ such that for $\boldsymbol{X} \sim \prod_{i=1}^n \mathcal{D} * \mathcal{N}(0, \sigma_i^2 I)$:*

$$\mathbb{P}_{\boldsymbol{X}} \left\{ W_1(\widehat{T}(\boldsymbol{X}, \mathcal{D})) \geqslant c \max \left\{ \left( \frac{d + \log(1/\delta)}{\sum_{i=1}^n 1/\sigma_i^4} \right)^{1/4}, \frac{1}{k} \left( \frac{1}{\log(1/\delta)} \cdot \sum_{i=1}^n \frac{1}{\sigma_i^{4k-2}} \right)^{-1/(4k-2)} \right\} \right\} \geqslant \delta.$$

In the remainder of the section, we present a proof of the upper bound (Theorem 4.2) in the simplified setting of $d = 1$.

## 4.2 PROOF: ONE-DIMENSIONAL SETTING

We draw upon the approach of Wu & Yang (2020) for the homogeneous setting. We start by recalling some notation from Wu & Yang (2020). For $l \in \mathbb{N}$ and distribution, $\mathcal{D}$, over $\mathbb{R}$:

$$m_l(\mathcal{D}) := \mathbb{E}_{X \sim \mathcal{D}}[X^l] \quad \text{and} \quad \mathbf{m}_l(\mathcal{D}) := [m_0(\mathcal{D}), m_1(\mathcal{D}), \dots, m_l(\mathcal{D})].$$

The estimator of Wu & Yang (2020) utilizes the Hermite polynomials defined as: $H_r(x) = r! \sum_{j=0}^{\lfloor r/2 \rfloor} \frac{(-1/2)^j}{j!(r-2j)!} x^{r-2j}$. These polynomials satisfy the following crucial denoising property: $\mathbb{E}_{X \sim \mathcal{N}(\mu,1)}[H_r(X)] = \mu^r$. Linearity of expectation now allows for denoising general *convolutions* with Gaussians:

$$\gamma_{r,\sigma}(x) := \sigma^r H_r(x/\sigma) = r! \sum_{j=0}^{\lfloor r/2 \rfloor} \frac{(-1/2)^j}{j!(r-2j)!} \sigma^{2j} x^{r-2j} \text{ satisfies } \mathbb{E}_{X \sim \mathcal{D} * \mathcal{N}(\mu,\sigma^2)}[\gamma_{r,\sigma}(X)] = E_{X \sim \mathcal{D}}[X^r].$$

Hence, they allow approximate recovery of the moments of the $k$-atomic distribution that was convolved with a Gaussian to produce the final mixture. The main issue with this approach is that the moments are only recovered *approximately*. The following technical result, implicit in Wu & Yang (2020) and proved in Appendix B, shows that error in estimating the moments can be directly translated to an error in Wasserstein distance in an estimation procedure.

**Lemma 4.1.** *(Wu & Yang (2020)).* Let $k \in \mathbb{N}$, $\delta \in [0, 1]$. Let $\mathcal{D}$ be a $k$-atomic distribution supported on $[-1, 1]$ and $\widetilde{m}_1, \dots, \widetilde{m}_{2k-1}$ be estimates satisfying $|\widetilde{m}_i - m_i(\mathcal{D})| \leqslant \delta$ for all $i \in [2k-1]$. Then there is a procedure running in time $O_k(1)$, which on input $\{\widetilde{m}_i\}_{i=1}^{2k-1}$, returns distribution $\widehat{\mathcal{D}}$ satisfying:

$$W_1(\mathcal{D}, \widehat{\mathcal{D}}) \leqslant O(k\delta^{\frac{1}{2k-1}}).$$

Finally, the error in the moment estimates can be bounded by the variance of the Hermite polynomials.

**Lemma 4.5** (Wu & Yang (2020)). *Let $M > 0$ and $X \sim \mathcal{D} * \mathcal{N}(0, \sigma^2)$ with $\mathcal{D}$ supported on $[-M, M]$. Then,*

$$\forall r \in \mathbb{N} : \text{Var}(\gamma_{r,\sigma}(X)) \leqslant (O(M + \sigma\sqrt{r}))^{2r}.$$

Lemma 4.5 and Lemma 4.1 now reduce the problem of estimating the parameters of the mixture to that of estimating its first $2k - 1$ moments. Our algorithm is presented in Algorithm 3. The key step in the algorithm is the addition of the weights, $\alpha_i$, which down-weight samples with large variance. These weights are chosen to minimize the variance of obtained moment estimates. Intuitively, samples with large variance are less reliable and their contributions discounted. Interestingly, the degree of down weighting also scales with the *complexity* of the distribution, measured by $k$. Next, we formally establish the correctness of Algorithm 3 through the following simplified theorem.

**Theorem 4.6.** *Let $M > 1$ be a constant and $k, n \in \mathbb{N}$. Suppose $\mathcal{D}$ is a $k$-atomic distribution supported on $[-M, M]$ and that $\boldsymbol{X} = \{X_i\}_{i=1}^n$ are generated as $X_i \sim \mathcal{D} * \mathcal{N}(0, \sigma_i^2)$ with $\sigma_i \geqslant 1$. Then, Algorithm 3 given as input $\boldsymbol{X}$ and $\{\sigma_i\}_{i \in [n]}$ returns with probability at least $3/4$ a $k$-atomic distribution $\widehat{\mathcal{D}}$ satisfying:*

$$W_1(\mathcal{D}, \widehat{\mathcal{D}}) \leqslant Ck^2 \cdot \left( \sum_{i=1}^n \frac{1}{\sigma_i^{4k-2}} \right)^{-1/(4k-2)}$$

*Proof.* We start by rescaling the distribution so that $M = 1$. Next, we get from Lemma 4.5 that for all $r \in [2k-1]$:

$$\text{Var}(\widetilde{m}_r) = \sum_{i=1}^{n} \alpha_i^2 \text{Var}\left(\gamma_{r,\sigma_i}(X_i)\right) \leqslant \sum_{i=1}^{n} \alpha_i^2 (C\sqrt{k}\sigma_i)^{4k-2} = (C\sqrt{k})^{4k-2} \cdot \left(\sum_{i=1}^{n} \frac{1}{\sigma_i^{4k-2}}\right)^{-1} =: \sigma_m^2.$$

Therefore, we get by a consequence of Chebyshev's inequality for any $r \in [2k-1]$:

$$|\widetilde{m}_r - m_r| \leqslant 4\sqrt{k}\sigma_m$$

with probability at least $1 - 1/16k$. Hence, we get by the union bound:

$$\forall r \in [2k-1] : |\widetilde{m}_r - m_r| \leqslant 4\sqrt{k}\sigma_m$$

with probability at least $3/4$. Conditioned on this event, Lemma 4.1 concludes the proof of the theorem. □

The proof of the high-dimensional setting is presented in Appendix C. Algorithmically, the high-dimensional setting uses *two* different weighting schemes; one for computing the appropriate low-dimensional subspace that is *independent* of $k$ and another for an exhaustive search algorithm similar to Doss et al. (2023). Furthermore, our algorithm and analysis diverge from and refine the results of Doss et al. (2023). Consequently, the dependence on the failure probability improved from $\sqrt{\log(1/\delta)}$ to the *optimal* $(\log(1/\delta))^{1/(4k-2)}$ dependence. Our key technical improvement is due to a robustness analysis on the approximation of the high-dimensional Wasserstein distance by one-dimensional projections (see Appendix C.2). This allows bypassing the chaining-based arguments of Doss et al. (2023) and instead employing a median-of-means based approach with a simple covering number argument.

## 5 EXPERIMENTS

### 5.1 TRAINING ONLY WITH CORRUPTED DATA

We start our experimental section by presenting results in the setting of prior work (Daras et al., 2024; 2023b), i.e. using only noisy data. We train models at three different levels of corruption: i) $\sigma = 0.05$, ii) $\sigma = 0.1$, iii) $\sigma = 0.2$, across three different datasets of different sizes: CIFAR-10 (60K samples), CelebA-HQ (30K samples) and ImageNet ($\approx 1.3$M images). We underline that prior work (Daras et al., 2024) only finetuned with noisy data. Instead, we opt for training from scratch which allows us to measure different quantities of interest in a more controlled setting.

For reference, we visualize the corruption levels in Figure 3. We train models with and without consistency. For our experiments, training with consistency was $2-3$x slower and used 2x GPUs. The full experimental results are provided in Appendix F. For the models without consistency, we report results for both truncated (see Algorithm 2) and full sampling. For the models trained with consistency, we report results for the full sampling algorithm, as consistency is supposed to help mostly with times $t < t_n$. The results of this experiment are summarized in Table 2. The findings are the following: **i)** the results are very poor for a model trained without consistency when the full-sampling algorithm is used, i.e. the model cannot extrapolate to times $t < t_n$, if it was only trained for times $t \geqslant t_n$, **ii)** Truncated Sampling helps, but still, even for low corruptions, (e.g. for $\sigma = 0.05$) the performance is relatively weak, and, **iii)** training with consistency is beneficial, particularly for $\sigma = 0.05$ and $\sigma = 0.1$, but as the corruption increases the benefits of it are shrinking. Another interesting observation is that the performance deteriorates less (for the same corruption level) compared to the clean performance as the dataset size grows. This is expected: in the limit of infinite data, models trained with solely noisy data can still, in theory, perfectly recover the distribution (Daras et al., 2024). That said, even for ImageNet level datasets, at high corruption the best performance, obtained by training with the expensive consistency loss, is significantly inferior to the performance obtained by training with clean data.

### 5.2 TRAINING WITH A MIXTURE OF CLEAN AND NOISY DATA

The previous experiments showed that training with solely highly noisy data leads to a significant performance deterioration, at least for datasets with sizes up to ImageNet, and this also comes at a significant computational cost because of the consistency loss. But what if a small set of clean images is available? Arguably, this setting is fairly realistic since in the dataset design process one can spend some initial effort (measured in money, computation, or otherwise) to collect some high-quality data points and then rely on cheaper noisy points for the rest of the dataset. To the best of our knowledge, this setting has not been explored by prior work. To study this, we fix the dataset size and we change

| Dataset | Noise Level | Full Sampling | Truncated Sampling | Consistency + Full Sampling |
|---|---|---|---|---|
| CIFAR-10 | 0.00 | $1.99 \pm_{0.02}$ | NA | NA |
| | 0.05 | $8.78 \pm_{0.02}$ | $4.53 \pm_{0.08}$ | $2.82 \pm_{0.02}$ |
| | 0.10 | $25.55 \pm_{0.10}$ | $7.55 \pm_{0.07}$ | $3.63 \pm_{0.03}$ |
| | 0.20 | $60.73 \pm_{0.21}$ | $12.12 \pm_{0.03}$ | $11.93 \pm_{0.09}$ |
| CelebA-HQ | 0.00 | $2.40 \pm_{0.07}$ | NA | NA |
| | 0.05 | $12.77 \pm_{0.02}$ | $6.73 \pm_{0.02}$ | $5.50 \pm_{0.03}$ |
| | 0.10 | $45.90 \pm_{0.08}$ | $10.27 \pm_{0.01}$ | $9.38 \pm_{0.02}$ |
| | 0.20 | $61.14 \pm_{0.14}$ | $13.90 \pm_{0.01}$ | $12.97 \pm_{0.11}$ |
| ImageNet | 0.00 | $1.41 \pm_{0.03}$ | NA | NA |
| | 0.05 | $2.39 \pm_{0.01}$ | $2.53 \pm_{0.01}$ | $2.26 \pm_{0.01}$ |
| | 0.10 | $6.23 \pm_{0.01}$ | $3.67 \pm_{0.01}$ | $2.99 \pm_{0.05}$ |
| | 0.20 | $18.08 \pm_{0.05}$ | $6.03 \pm_{0.02}$ | $5.32 \pm_{0.08}$ |

Table 2: **FID** comparisons between models trained on solely noisy data across datasets and noise levels. In the first two columns, we report results without consistency training, with and without sampling truncation. Training with consistency (third col.) significantly improves the results, but the performance is still poor compared to the clean data training for high corruption.

the percentage $p\%$ of clean data. For every value of $p \in \{0.0, 0.1, 0.3, 0.5, 0.7, 0.9, 1.0\}$, we train models with: i) $p\%$ clean data, and, ii) with $p\%$ clean data and $(1-p)\%$ of noisy data at different levels of noise. To train with both clean and noisy data we used the method described in Section 3 and in Algorithm 1. We underline that for these experiments we do **not** use consistency loss and hence the training cost stays the same as training with clean data.

We report our results in Table 3 and we highlight a subset of the results in Table 1. The results show that even when we have 90% noisy data and high-corruption, i.e. $\sigma = 0.2$, the performance stays near the state-of-the-art as long as a small set (10%) of clean data is available. In fact, the obtained performance is far superior to the performance obtained by training only with clean data and the performance obtained by training with only noisy data, indicating that the benefit comes from the mixture of the two. The difference is quite dramatic: i) 10% clean ImageNet samples $\Rightarrow 10.57$ FID, ii) 100% noisy samples (at $\sigma = 0.2$) $\Rightarrow 5.32$ FID, iii) 90% noisy samples and 10% clean samples $\Rightarrow$ **1.68** FID. This phenomenon is further illustrated in Figure 1.

Table 3: **FID** comparisons as we change the mix of clean and noisy data. Data availability is reported as percentage of the total dataset size: 60K for CIFAR, 30K for CelebA-HQ and $\approx$ 1.3M for ImageNet. Training with a mix of clean and noisy data is significantly better than training purely on a few clean data points or purely on the full noisy dataset.

| Dataset | Available Data | Noise Level | p% of clean data | | | | | | |
|---|---|---|---|---|---|---|---|---|---|
| | | | 100% | 90% | 70% | 50% | 30% | 10% | 0% |
| CIFAR-10 | p% clean | N/A | | $2.07_{\pm0.01}$ | $2.20_{\pm0.03}$ | $2.40_{\pm0.02}$ | $3.99_{\pm0.01}$ | $17.30_{\pm0.14}$ | N/A |
| | p% clean, (1-p)% noisy | 0.05 | $1.99_{\pm0.02}$ | $2.04_{\pm0.02}$ | $2.04_{\pm0.03}$ | $2.06_{\pm0.00}$ | $2.11_{\pm0.02}$ | $2.17_{\pm0.04}$ | $8.78_{\pm0.02}$ |
| | | 0.10 | | $2.06_{\pm0.03}$ | $2.14_{\pm0.03}$ | $2.15_{\pm0.01}$ | $2.24_{\pm0.02}$ | $2.34_{\pm0.03}$ | $25.55_{\pm0.10}$ |
| | | 0.20 | | $2.06_{\pm0.03}$ | $2.14_{\pm0.02}$ | $2.24_{\pm0.01}$ | $2.42_{\pm0.03}$ | $2.81_{\pm0.02}$ | $60.73_{\pm0.21}$ |
| CelebA-HQ | p% clean | N/A | | $2.50_{\pm0.01}$ | $2.68_{\pm0.00}$ | $2.83_{\pm0.01}$ | $3.72_{\pm0.01}$ | $11.92_{\pm0.05}$ | N/A |
| | p% clean, (1-p)% noisy | 0.05 | $2.40_{\pm0.07}$ | $2.40_{\pm0.01}$ | $2.45_{\pm0.01}$ | $2.45_{\pm0.02}$ | $2.50_{\pm0.01}$ | $2.50_{\pm0.02}$ | $12.77_{\pm0.02}$ |
| | | 0.10 | | $2.40_{\pm0.01}$ | $2.48_{\pm0.02}$ | $2.51_{\pm0.04}$ | $2.51_{\pm0.01}$ | $2.67_{\pm0.01}$ | $45.90_{\pm0.10}$ |
| | | 0.20 | | $2.50_{\pm0.00}$ | $2.51_{\pm0.11}$ | $2.52_{\pm0.01}$ | $2.67_{\pm0.02}$ | $2.75_{\pm0.02}$ | $61.14_{\pm0.14}$ |
| ImageNet | p% clean | N/A | | $1.50_{\pm0.02}$ | $1.65_{\pm0.01}$ | $2.15_{\pm0.01}$ | $4.34_{\pm0.06}$ | $10.57_{\pm0.06}$ | N/A |
| | p% clean, (1-p)% noisy | 0.05 | $1.41_{\pm0.03}$ | $1.46_{\pm0.02}$ | $1.46_{\pm0.02}$ | $1.46_{\pm0.02}$ | $1.47_{\pm0.02}$ | $1.51_{\pm0.02}$ | $2.40_{\pm0.02}$ |
| | | 0.10 | | $1.48_{\pm0.01}$ | $1.48_{\pm0.02}$ | $1.49_{\pm0.01}$ | $1.49_{\pm0.01}$ | $1.57_{\pm0.02}$ | $6.23_{\pm0.01}$ |
| | | 0.20 | | $1.50_{\pm0.02}$ | $1.51_{\pm0.02}$ | $1.51_{\pm0.02}$ | $1.59_{\pm0.03}$ | $1.68_{\pm0.02}$ | $18.08_{\pm0.05}$ |

We test the limits of our method by evaluating it in extreme corruption, either in terms of number of clean samples or in terms of the amount of noise. For computational reasons, we only present these results for CIFAR-10.

**Higher corruption.** We use 90% noisy and 10% clean data and we increase the corruption to $\sigma = 0.4$. The FID becomes $\mathbf{3.56 \pm 0.03}$, which is still much better than the performance (17.30) obtained by using 10% of clean data.

**Less clean data.** We fix $\sigma = 0.2$ and we severely decrease the number of clean data. Specifically, we evaluate performance when 1% of clean data and 99% of noisy data are available. The performance of this model is $\mathbf{3.53 \pm 0.03}$. For

reference, the performance of the model trained with $100\%$ noisy data at this noise level is $60.73$ without Truncated Sampling and $11.93$ with consistency training. Hence, $1\%$ of clean data has a tremendous effect in performance.

### 5.3 How much is a noisy image worth?

*How much exactly is a noisy image worth?* In what follows, we address the question of **data pricing**. Given finite resources for data curation, how should we allocate them to obtain the best performance? On the flip side, what is a fair pricing scheme that reflects data utility? We approach this through the notion of **effective sample size** suggested by our theoretical model. Let:

$$n_d = \frac{\sum_{i=1}^n 1/\sigma_i^4}{\max_i 1/\sigma_i^4} \quad n_l = \frac{\sum_{i=1}^n 1/\sigma_i^{4k-2}}{\max_i 1/\sigma_i^{4k-2}},$$

be the two terms that appear in the bound of Theorem 4.2. Assume that we operate in the asymptotic setting with large $n$ and with the fraction of clean and noisy data bounded away from $0$ by a constant independent of $n$. In this regime, the dominant term in Theorem 4.2 is the second due to its slower rate of convergence. Therefore, the effective sample size for low-dimensional estimation, $n_l$, is the main determinant of the error.

This implies that we can use a fixed pricing scheme, largely independent of the sample size $n$. To validate this theoretical prediction, let's price clean samples at $1\$$ per unit and noisy samples at $c_\sigma\$$ per unit, for a parameter $c_\sigma$ that we will bound using data from our experiments. Specifically, to bound $c_\sigma$ we can derive inequalities by comparing performances from Table 3. For example, the CIFAR-10 performance using $90\%$ clean data is $2.07$ which is better than the $2.42$ FID performance obtained using $30\%$ clean and $70\%$ noisy samples at $\sigma = 0.2$, which implies that:

$$0.9n \geqslant 0.3n + c_{0.2}0.7n \implies c_{0.2} \leqslant \frac{6}{7}.$$

We repeat this computation for all pairs of the table and derive more inequalities. We restrict comparisons to settings with at least $10\%$ clean samples to ensure we remain in the asymptotic regime. The best upper and lower bounds are:

$$\text{CIFAR-10, CelebA:} \quad \boxed{1.17 \leqslant \frac{1}{c_{0.05}} \leqslant 1.25}, \quad \text{ImageNet:} \quad \boxed{1.125 \leqslant \frac{1}{c_{0.05}} \leqslant 1.17}$$

$$\text{CIFAR-10, CelebA:} \quad \boxed{1.25 \leqslant \frac{1}{c_{0.1}} \leqslant 1.5}, \quad \text{ImageNet:} \quad \boxed{1.125 \leqslant \frac{1}{c_{0.1}} \leqslant 1.17}$$

$$\text{CIFAR-10, CelebA, ImageNet:} \quad \boxed{1.5 \leqslant \frac{1}{c_{0.2}} \leqslant 1.75}$$

The quantity $1/c_\sigma$ measures the number of noisy samples required to compensate for a clean one. In line with intuition, for larger values of $\sigma$ (i.e. the noisier the sample), more noisy samples required to replace a clean one. In addition, this ratio is the lowest for the largest dataset, ImageNet. The narrow ranges of the upper and lower bounds, lend credibility to this data pricing approach. Finer-grid evaluation at Table 3 would allow for even narrower estimates.

## 6 Limitations and Future Work

In this paper, we only studied the case of additive Gaussian Noise corruption. In practice, the corruption might be more complex or even unknown. We further focused on the scale of having two noise levels, i.e. clean data and noisy data. The approach can be naturally extended to arbitrary many noise levels. Further, all the results in this work were developed for pixel-space diffusion models. A promising avenue for future research is to extend these findings to latent diffusion models. Finally, due to the high computational training requirements, we were only able to scale up to ImageNet and $64 \times 64$ resolution. It is worth exploring how these behaviors change as we further increase the dimension and the dataset size. On the theory side, our Gaussian Mixtures Model might not capture all the intricacies of real distributions and while being a useful tool, we should always interpret these results with caution.

### Acknowledgments

This research has been supported by NSF Grants AF 1901292, CNS 2148141, Tripods CCF 1934932, IFML CCF 2019844 and research gifts by Western Digital, Amazon, WNCG IAP, UT Austin Machine Learning Lab (MLL),

Cisco and the Stanly P. Finch Centennial Professorship in Engineering. Constantinos Daskalakis has been supported by NSF Awards CCF-1901292, DMS-2022448 and DMS-2134108, a Simons Investigator Award, and the Simons Collaboration on the Theory of Algorithmic Fairness. Giannis Daras has been supported by the Onassis Fellowship (Scholarship ID: F ZS 012-1/2022-2023), the Bodossaki Fellowship and the Leventis Fellowship.

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

## A APPENDIX

---

**Algorithm 2** Truncated Sampling Sampling Algorithm.

---

**Input:** network $\boldsymbol{h}_\theta$, time steps $T$, step size $\Delta t$, noise schedule $\{\sigma_t\}_{t=0}^T$, stopping time $\sigma_{t_n}$.

1: $\boldsymbol{x}_T \sim \mathcal{N}(0, \sigma_T)$          ▷ Initialization
2: **for** $t = T, T-1, \ldots, 1$ **do**
3:     $\hat{\boldsymbol{x}}_0 \leftarrow \boldsymbol{h}_\theta(\boldsymbol{x}_t, t)$
4:     **if** $\sigma_{t-1} < \sigma_{t_n}$ **then**
5:        **return** $\hat{\boldsymbol{x}}_0$          ▷ Truncated Sampling
6:     **end if**
7:     $\boldsymbol{x}_{t-1} \leftarrow x_t - \frac{\sigma_t - \sigma_{t-1}}{\sigma_t}(\boldsymbol{x}_t - \hat{\boldsymbol{x}}_0)$
8: **end for**
9: **return** $x_0$

---

## B PROOF OF LEMMA 4.1

In this section, we establish Lemma 4.1 used to prove the weakened bound for the one-dimensional setting in Theorem 4.6. This result is implicit in Wu & Yang (2020) and we include the proof here for completeness. Before we proceed, we reproduce the following result from Wu & Yang (2020) which shows that for two distributions with finite support and close moments, their Wasserstein distance is also close.

**Proposition B.1.** *Let $\mathcal{D}$ and $\mathcal{D}'$ be $k$-atomic distributions supported on $[-1, 1]$. If $|m_i(\mathcal{D}) - m_i(\mathcal{D}')| \leqslant \delta$ for $i = 1, \ldots, 2k - 1$, then:*

$$\mathrm{W}_1(\mathcal{D}, \mathcal{D}') \leqslant O(k\delta^{\frac{1}{2k-1}}).$$

---

**Algorithm 3** Denoised Method of Moments with Heterogenous Variances

---

**Input:** Samples $X_i \sim \mathcal{M}_{\text{Dirac}} * \mathcal{N}(0, \sigma_i^2)$
**Output:** Candidate $k$-mixture $\hat{\mathcal{M}}_{\text{Dirac}} = \sum_{i=1}^{k} \hat{w}_i \delta_{\hat{x}_i}$
1: Define weights $\alpha_i$, for $i \in [n]$:

$$\alpha_i = \frac{1/\sigma_i^{4k-2}}{\sum_{j=1}^{n} 1/\sigma_j^{4k-2}}.$$

2: Define moment approx. for $r \in [2k-1]$:

$$\tilde{m}_r = \sum_{i=1}^{n} \alpha_i \gamma_{r,\sigma_i}(X_i).$$

3: Return output of Lemma 4.1 on input $\hat{\mathbf{m}}$.

---

We reproduce the statement of Lemma 4.1 and include its proof below.

**Lemma 4.1.** *(Wu & Yang (2020)). Let $k \in \mathbb{N}$, $\delta \in [0, 1]$. Let $\mathcal{D}$ be a $k$-atomic distribution supported on $[-1, 1]$ and $\tilde{m}_1, \ldots, \tilde{m}_{2k-1}$ be estimates satisfying $|\tilde{m}_i - m_i(\mathcal{D})| \leqslant \delta$ for all $i \in [2k-1]$. Then there is a procedure running in time $O_k(1)$, which on input $\{\tilde{m}_i\}_{i=1}^{2k-1}$, returns distribution $\widehat{\mathcal{D}}$ satisfying:*

$$\mathrm{W}_1(\mathcal{D}, \widehat{\mathcal{D}}) \leqslant O(k\delta^{\frac{1}{2k-1}}).$$

*Proof.* We recall some preliminaries from Wu & Yang (2020). First, the moment space defined for a set $K \subset \mathbb{R}$:

$$\mathcal{M}_r(K) = \{\mathbf{m}_r(\pi) : \pi \text{ is supported on } K\}.$$

Specifically when $K = [a, b]$, $\mathcal{M}_r$ is fully characterized by the following convex constraints:

$$\mathbf{M}_{0,r} \succcurlyeq 0, \quad (a+b)\mathbf{M}_{1,r-1} \succcurlyeq ab\mathbf{M}_{0,r-2} + \mathbf{M}_{2,r} \text{ for even } r$$
$$b\mathbf{M}_{0,r-1} \succcurlyeq \mathbf{M}_{1,r} \succcurlyeq \mathbf{M}_{0,r-1} \quad \text{for odd } r, \tag{MOM-CHAR}$$

where the matrices $\mathbf{M}_{i,j}$ are defined as follows for $m \in \mathcal{M}_r$:

$$\mathbf{M}_{i,j} = \begin{bmatrix} m_i & m_{i+1} & \cdots & m_{\frac{i+j}{2}} \\ m_{i+1} & m_{i+2} & \cdots & m_{\frac{i+j}{2}+1} \\ \vdots & \vdots & \ddots & \vdots \\ m_{\frac{i+j}{2}} & m_{\frac{i+j}{2}+1} & \cdots & m_j \end{bmatrix}.$$

As noted in Wu & Yang (2020), these constraints are easy to see as necessary. Surprisingly, they also turn out be *sufficient* for characterizing the space $\mathcal{M}_r$ Shohat & Tamarkin (1943).

The algorithm now establishing Lemma 4.1 is presented in Algorithm 4. By our assumptions on the input, we have:

$$\forall i \in [2k-1] : |\widehat{m}_i - \tilde{m}_i| \leqslant \delta \implies |\widehat{m}_i - m_i(\mathcal{D})| \leqslant \delta$$

where the first inequality follows from the fact that $\widehat{m}$ is the minimizer of the convex program and $\mathcal{D}$ certifies that a solution with small error exists and the second follows from the triangle inequality. Next, since $\widehat{\mathbf{m}}$ corresponds to the moments of a valid distribution from the sufficiency criterion for one-dimensional distributions, Algorithm 5 returns a $k$ atomic distribution matching the moments $\widehat{\mathbf{m}}$. An application of Proposition B.1 establishes the lemma. $\square$

## C   HIGH-DIMENSIONAL UPPER BOUND: PROOF OF THEOREM 4.2

In this section, we prove Theorem 4.2. Here, we adopt the techniques from Doss et al. (2023) which reduces the high dimensional setting to the one-dimensional setting through a series of reductions. The first reduces the $d$-dimensional setting to the $k$-dimensional setting by a projection onto the principal components of the data. This is a standard

---

**Algorithm 4** Estimate Distribution from Moments

---

**Input:** Moment estimates $\widetilde{m}_1, \ldots, \widetilde{m}_{2k-1}$
**Output:** Candidate $k$-mixture $\widehat{\mathcal{M}}_{\text{Dirac}} = \sum_{i=1}^k \widehat{w}_i \delta_{\widehat{x}_i}$
  1: Let $\widehat{\mathbf{m}}$ be the solution to the following:

$$\min \{\|\widetilde{\mathbf{m}} - \widehat{\mathbf{m}}\| : \widehat{\mathbf{m}} \text{ satisfies MOM-CHAR}\} \text{ where } \widetilde{\mathbf{m}} = (\widetilde{m}_1, \ldots, \widetilde{m}_{2k-1}).$$

  2: Return the output of Algorithm 5 on input $\widehat{\mathbf{m}}$.

---

**Algorithm 5** Gauss Quadrature

---

**Input:** Valid moment vector $\mathbf{m} = (m_1, \ldots, m_{2k-1})$
**Output:** Quadrate points $(x_1, \ldots, x_k)$ and weights $(w_1, \ldots, w_k)$
  1: Define the degree-$k$ polynomial:

$$P(x) = \begin{bmatrix} 1 & m_1 & \ldots & m_k \\ \vdots & \vdots & \ddots & \vdots \\ m_{k-1} & m_k & \ldots & m_{2k-1} \\ 1 & x & \ldots & x^k \end{bmatrix}.$$

  2: Now, define $(x_1, \ldots, x_k)$ as the roots of $P(x)$
  3: The weights $(w_1, \ldots, w_k)$ are defined as:

$$w = \begin{bmatrix} 1 & 1 & \ldots & 1 \\ x_1 & x_2 & \ldots & x_k \\ \vdots & \vdots & \ddots & \vdots \\ x_1^{k-1} & x_2^{k-1} & \ldots & x_k^{k-1} \end{bmatrix}^{-1} \begin{bmatrix} 1 \\ m_1 \\ \vdots \\ m_{k-1} \end{bmatrix}.$$

---

procedure that is frequently employed to reduce high-dimensional estimation to their more tractable low-dimensional counterparts. The second step follows by establishing a result which shows that *any* candidate mixture whose *one-dimensional marginals* are approximately consistent with the original mixture is close to the original mixture in Wasserstein distance. In the remainder of the section, Appendix C.1 analyzes the low-dimensional projection step, Appendix C.2 presents the low-dimensional estimation algorithm and finally, Appendix C.3 establishes Theorem 4.2.

## C.1 REDUCTION TO LOW-DIMENSIONS

Here, we establish concentration bounds on the matrix, $\widehat{\Sigma}$, used to compute the low-dimensional projection in Algorithm 8. We recall the following result that we will repeatedly use in our proofs:

**Lemma C.1** (Exercise 4.4.3 Vershynin (2018)). *Let $d \in \mathbb{N}$ and $\mathcal{G}$ be an $\varepsilon$-net of the unit sphere. Then, we have for any symmetric $M \in \mathbb{R}^{d \times d}$:*

$$\|M\| \leqslant \frac{1}{1 - \varepsilon} \max_{v \in \mathcal{G}} |v^\top M v|.$$

We first bound the noise terms that arise when computing the high-dimensional covariance.

**Lemma C.2.** *Let $g_1, \ldots, g_n \overset{i.i.d}{\sim} \mathcal{N}(0, I)$ and $\alpha_i \geqslant 0$ for $i \in [n]$ with $\sum_{i=1}^n \alpha_i = 1$. Then, we have:*

$$\left\|\sum_{i=1}^n \alpha_i g_i g_i^\top - I\right\| \leqslant \max\left(\sqrt{(d + \log(1/\delta)) \sum_{i=1}^n \alpha_i^2}, (d + \log(1/\delta)) \max_i \alpha_i\right)$$

*with probability at least $1 - \delta$.*

*Proof.* As is standard, we note the following characterization of the spectral norm for any symmetric $M \in \mathbb{R}^{d \times d}$:

$$\|M\| = \max_{\|v\|=1} |v^\top M v|$$

and start with a fixed $v$. The extension to the remainder of values is obtained through a discretization argument, a net over the unit sphere and a union bound. Now, for fixed $\|v\| = 1$, consider the random variable:

$$W_v := v^\top \left( \sum_{i=1}^n \alpha_i g_i g_i^\top - I \right) v = \sum_{i=1}^n \alpha_i (\langle g_i, v \rangle^2 - 1).$$

We now prove that $W$ is sub-exponential:

**Claim C.3.** We have for any $|\lambda| < 1/4$ and $g \sim \mathcal{N}(0,1)$:

$$\mathbb{E}[\exp \lambda(g^2 - 1)] \leqslant \exp(4\lambda^2).$$

*Proof.* We have for any $|\lambda| < 1/4$ and $g \sim \mathcal{N}(0,1)$:

$$
\begin{aligned}
\mathbb{E}[\exp(\lambda(g^2 - 1))] &= e^{-\lambda} \int_{-\infty}^\infty \frac{1}{\sqrt{2\pi}} \exp\left\{ (\lambda g^2) - \frac{g^2}{2} \right\} dg \\
&= e^{-\lambda} \int_{\infty}^\infty \frac{1}{\sqrt{2\pi}} \exp\left\{ -\frac{(1 - 2\lambda)}{2} \cdot g^2 \right\} dg \\
&= e^{-\lambda} \cdot \frac{1}{\sqrt{1 - 2\lambda}} = e^{-\lambda} \cdot \frac{\sqrt{1 + 2\lambda}}{\sqrt{1 - 4\lambda^2}} \\
&\leqslant e^{-\lambda} \cdot \frac{e^\lambda}{\sqrt{1 - 4\lambda^2}} \leqslant \frac{1}{\sqrt{e^{-8\lambda^2}}} = \exp(4\lambda^2)
\end{aligned}
$$

concluding the proof of the claim. $\qquad\square$

Hence, we have that each of the $(\langle g_i, v \rangle^2 - 1)$ are $C$-sub-exponential random variables for some constant $C > 0$. Hence, we have by Bernstein's inequality (Vershynin, 2018, Theorem 2.8.2):

$$\mathbb{P}\{W_v \geqslant t\} \leqslant 2 \exp\left( -c \min\left( \frac{t^2}{\sum_{i=1}^n \alpha_i^2}, \frac{t}{\max_i \alpha_i} \right) \right).$$

Now, let $\mathcal{Q}$ be a $1/4$-cover of the set $\{x : \|x\| = 1\}$ with $|Q| \leqslant 9^d$. Then, we have by the union bound:

$$\mathbb{P}\{\exists v \in Q : W_v \geqslant t\} \leqslant 2 \cdot 9^d \exp\left( -c \min\left( \frac{t^2}{\sum_{i=1}^n \alpha_i^2}, \frac{t}{\max_i \alpha_i} \right) \right).$$

Choosing $t^*$ as:

$$t^* = \max\left( \sqrt{(d + \log(1/\delta)) \sum_{i=1}^n \alpha_i^2}, (d + \log(1/\delta)) \max_i \alpha_i \right)$$

yields:

$$\max_{v \in Q} W_v \leqslant t^*$$

with probability at least $1 - \delta$. Observing that:

$$\|M\| = \max_{\|v\|=1} |v^\top M v| \leqslant 2 \max_{v \in \mathcal{Q}} |v^\top M v|$$

for any symmetric matrix $M$ (Lemma C.1), concludes the proof. $\qquad\square$

Now, we formally establish concentration of the second moment matrix $\widehat{\Sigma}$, in spectral norm.

**Lemma C.4.** *Let $\mathcal{D}$ be a $k$-atomic distribution supported such that each element, $x$, in its support satisfies $\|x\| \leqslant R$. Furthermore, suppose $X_1, \ldots, X_n$ are independent samples with $X_i \sim \mathcal{D} * \mathcal{N}(0, \sigma_i^2 I)$ with $\sigma_i^2 > 0$ for all $i \in [n]$. Then, we have:*

$$\left\| \sum_{i=1}^n \alpha_i X_i X_i^\top - \bar{\sigma}^2 I - \mathbb{E}_{X \sim \mathcal{D}}[XX^\top] \right\| \leqslant$$

$$\max\left(\sqrt{(d + \log(1/\delta)) \sum_{i=1}^n \alpha_i^2 \sigma_i^2 \max(\sigma_i^2, R^2)}, (d + \log(1/\delta)) \max_i \alpha_i \sigma_i^2, R^2 k \sqrt{\log(k/\delta) \sum_{i=1}^n \alpha_i^2}\right)$$

*where* $\alpha_i = \frac{1/\sigma_i^4}{\sum_{j=1}^n 1/\sigma_j^4}$ *and* $\bar{\sigma}^2 = \sum_{i=1}^n \alpha_i \sigma_i^2$ *with probability at least* $1 - \delta$.

*Proof.* Note that we may write $X_i = Y_i + \sigma_i g_i$ where $Y_i \sim \mathcal{D}$ and $g_i \sim \mathcal{N}(0, I)$ independently. Now, we write the error term as follows:

$$\sum_{i=1}^n \alpha_i X_i X_i^\top - \bar{\sigma}^2 I - \mathbb{E}_{X \sim \mathcal{D}}[XX^\top]$$

$$= \left(\sum_{i=1}^n \alpha_i Y_i Y_i^\top - \mathbb{E}_{X \sim \mathcal{D}}[XX^\top]\right) + \left(\sum_{i=1}^n \alpha_i \sigma_i^2 g_i g_I^\top - \bar{\sigma}^2 I\right) + \sum_{i=1}^n \alpha_i \sigma_i (Y_i g_i^\top + g_i Y_i^\top).$$

The proof proceeds by bounding the errors of each of the three terms separately. We start with the middle term which we bound with Lemma C.2 to obtain the following:

$$\left\|\sum_{i=1}^n \alpha_i \sigma_i^2 g_i g_I^\top - \bar{\sigma}^2 I\right\| \leqslant \max\left(\sqrt{(d + \log(1/\delta)) \sum_{i=1}^n \alpha_i^2 \sigma_i^4}, (d + \log(1/\delta)) \max_i \alpha_i \sigma_i^2\right).$$

For the first term, note that the error matrix may be written as follows:

$$\sum_{i=1}^n \alpha_i Y_i Y_i^\top - \mathbb{E}_{X \sim \mathcal{D}}[XX^\top] = \sum_{i=1}^k (\widehat{w}_i - w_i)\mu_i \mu_i^\top$$

where $\widehat{w}_i := \sum_{j=1}^n \alpha_j \mathbf{1}\{Y_j = \mu_i\}$ corresponds to the empirical counterpart of the mixture weights. Now, we have for any $\|v\| = 1$:

$$\left|v^\top \left(\sum_{i=1}^n \alpha_i Y_i Y_i^\top - \mathbb{E}_{X \sim \mathcal{D}}[XX^\top]\right)v\right| \leqslant R^2 \sum_{i=1}^k |\widehat{w}_i - w_i|.$$

An application of Hoeffding's inequality and the union bound obtains:

$$\forall i \in [k] : |\widehat{w}_i - w_i| \leqslant C\sqrt{\log(k/\delta) \sum_{i=1}^n \alpha_i^2}$$

with probability at least $1 - \delta/4$. Conditioned on this event, we have:

$$\left\|\sum_{i=1}^n \alpha_i Y_i Y_i^\top - \mathbb{E}_{X \sim \mathcal{D}}[XX^\top]\right\| \leqslant CR^2 k \sqrt{\log(k/\delta) \sum_{i=1}^n \alpha_i^2}.$$

We now proceed to the final term. Consider a $1/4$-net of the unit sphere, $\mathcal{G}$. We have for any fixed $v \in \mathcal{G}$:

$$v^\top \left(\sum_{i=1}^n \alpha_i \sigma_i (Y_i g_i^\top + g_i Y_i^\top)\right)v = \sum_{i=1}^n 2\alpha_i \sigma_i \langle v, Y_i \rangle \langle v, g_i \rangle.$$

Noting that $\langle v, g_i \rangle$ is a standard Gaussian random variable and that $\|Y_i\| \leqslant R$, we get that each term in the sum is sub-Gaussian and as a consequence, we get by Hoeffding's inequality:

$$\left|v^\top \left(\sum_{i=1}^n \alpha_i \sigma_i (Y_i g_i^\top + g_i Y_i^\top)\right)v\right| \leqslant C\sqrt{\log(1/\delta') \sum_{i=1}^n \alpha_i^2 \sigma_i^2 R^2}$$

with probability at least $1 - \delta'$. Hence, we get by the union bound over all $v \in \mathcal{G}$ and the size of $\mathcal{G}$:

$$\forall v \in \mathcal{G} : \left|v^\top \left(\sum_{i=1}^n \alpha_i \sigma_i (Y_i g_i^\top + g_i Y_i^\top)\right)v\right| \leqslant C\sqrt{(d + \log(1/\delta)) \sum_{i=1}^n \alpha_i^2 \sigma_i^2 R^2}.$$

Finally, this yields the same bound on spectral norm from Lemma C.1. The three bounds previously proved now conclude the proof of the lemma by the triangle inequality. $\qquad\square$

## C.2 ESTIMATION IN LOW-DIMENSIONS

The following lemma is a generalization of a key technical lemma in Doss et al. (2023). This lemma establishes that the Wasserstein distance between two $k$-atomic distributions over $\mathbb{R}^d$ can be approximated through various one-dimensional projections. While the version in Doss et al. (2023) required the supremum in the conclusion to be taken over all vectors on the unit sphere, our result only requires a supremum over a *grid* of the unit sphere. Crucially, the size of the grid is independent of $n$ and hence, the final error. Our algorithmic approach benefits from this since to establish concentration over these one-dimensional projections, we may use more robust median-of-means based approaches to obtain these individual estimates as opposed to the more intricate chaining-based techniques of Doss et al. (2023) which necessarily incurs the sub-optimal $\sqrt{\log(1/\delta)}$ dependence features in their final bound. The chaining-based techniques in Doss et al. (2023) are required since the grid size resulting from the previous approach is *dependent on* $n$ and hence, the simple techniques we employ incur additional factors depending on $n$. Technically, our improvement arises from a robustness analysis of the randomized one-dimensional projection argument employed in the proof.

**Lemma C.5.** *Let $d, k \in \mathbb{N}$ and $\mathcal{G}$ be a $1/(32k^2\sqrt{d})$-net of $\mathbb{S}^{d-1}$. Then, for two $k$-atomic distributions, $\mathcal{D}$ and $\mathcal{D}'$:*

$$\sup_{v \in \mathcal{G}} W_1(\mathcal{D}_v, \mathcal{D}'_v) \leqslant W_1(\mathcal{D}, \mathcal{D}') \leqslant 16k^2\sqrt{d} \sup_{v \in \mathcal{G}} W_1(\mathcal{D}_v, \mathcal{D}'_v).$$

*Proof.* The lower bound follows from (Doss et al., 2023, Lemma 3.1). For the upper bound, we proceed as in (Doss et al., 2023, Lemma 3.1) starting with a simple probabilistic argument. However, we additionally analyze the robustness properties of their construction to account for the discretization in our bound. First, define the set of vectors:

$$Q := \left\{ \frac{x - y}{\|x - y\|} : x, y \in \{\mu_i\}_{i=1}^k \cup \{\mu'_i\}_{i=1}^k \text{ and } x \neq y \right\}.$$

Note that $Q$ is of size at most $k^2$. We follow now the probabilistic argument also utilized in Doss et al. (2023) to establish the existence of a vector $v \in \mathbb{S}^{d-1}$ which satisfies:

$$\forall u \in Q : \|u\| \leqslant Ck^2\sqrt{d}|\langle v, u \rangle|.$$

Now, draw $v$ from the uniform distribution on $\mathbb{S}^{d-1}$. For any $u \in Q$, we have by standard results for $t \leqslant 1/2$:

$$\mathbb{P}\{|\langle v, u \rangle| \leqslant t\} = \frac{\Gamma(d/2)}{\sqrt{\pi}\Gamma((d-1)/2)} \int_{-t}^{t} (1 - x^2)^{(d-3)/2} dx \leqslant 4t\sqrt{d}.$$

Then, we get via a union bound:

$$\forall u \in Q : 8k^2\sqrt{d}\langle v, u \rangle \geqslant 1$$

with probability at least $1/2$. Hence, we have for such a $v$:

$$\forall x, y \in \{\mu_i\}_{i=1}^k \cup \{\mu'_i\}_{i=1}^k : \|x - y\| \leqslant 8k^2\sqrt{d}|\langle v, x - y \rangle|.$$

Now, let $\widetilde{v}$ be the closest neighbor of $v$ in $\mathcal{G}$. We now have for any $u \in Q$:

$$|\langle \widetilde{v}, u \rangle| \geqslant |\langle v, u \rangle| - \|\widetilde{v} - v\| \geqslant \frac{1}{16k^2\sqrt{d}}.$$

For this $\widetilde{v}$, observe that each of the elements in $\{\mu_i\}_{i=1}^k \cup \{\mu'_i\}_{i=1}^k$ is mapped to a distinct real number. Hence, every coupling of $\mathcal{D}$ and $\mathcal{D}'$ correspond to a coupling on the projections of the distributions $\mathcal{D}_{\widetilde{v}}$ and $\mathcal{D}'_{\widetilde{v}}$ and vice-versa. Then, consider any coupling of $\mathcal{D}_{\widetilde{v}}$ and $\mathcal{D}'_{\widetilde{v}}$, $C(Y, Y')$ and its corresponding high-dimensional analogue, $C_d(X, X')$. Then, we have:

$$16k^2\sqrt{d}\mathbb{E}_{(Y,Y')\sim C}[\|Y - Y'\|] \geqslant \mathbb{E}_{(X,X')\sim C}[\|X - X'\|] \geqslant W_1(\mathcal{D}, \mathcal{D}').$$

By maximizing over all such couplings, we derive the conclusion of the lemma. $\square$

The key benefit of Lemma C.5 is that it only requires us to obtain an candidate distribution whose marginals are close in Wasserstein distance to the marginals along a subset of directions *independent* of $n$. In the remainder of the proof, we will assume that $d = k$ as we may project the data down to $k$ dimensions by computing the PCA of the centered covariance matrix. Before we adapt the analysis of Doss et al. (2023), we will prove a moment concentration result which will enable a tightening of the dependence on the failure probability. While Doss et al. (2023) obtain an upper bound with a $\sqrt{\log(1/\delta)}$ dependence, our bound will instead grow was $(\log(1/\delta))^{1/(4k-2)}$.

**Lemma C.6.** *Let $k, m, n \in \mathbb{N}$, $\delta \in (0, 1/2)$, and $R > 0$ be a constant. Suppose $\mathcal{D}$ be a $k$-atomic distribution over $\mathbb{R}^k$ with each $x$ in its support satisfying $\|x\| \leqslant R$. Then, there is a procedure which when given independent samples $X_1, \ldots, X_n$ distributed according to $X_i \sim \mathcal{D} * \mathcal{N}(0, \sigma_i^2 I)$ with $\sigma_i \geqslant 1$ and $\max_i 1/\sigma_i^{4k-2} \leqslant c \sum_{i=1}^n 1/\sigma_i^{4k-2}/(\log(1/\delta) + k\log(k))$, returns one-dimensional estimates, $\{\mathcal{D}_v\}_{v \in \mathcal{G}}$, satisfying:*

$$\forall v \in \mathcal{G} : W_1(\mathcal{D}_v, \widehat{\mathcal{D}}_v) \leqslant Ck^2 \cdot \left( \frac{1}{(k\log(k) + \log(1/\delta))} \cdot \sum_{i=1}^n \frac{1}{\sigma_i^{4k-2}} \right)^{-1/(4k-2)}$$

*where $\mathcal{G}$ is a $(1/\operatorname{poly}(k))$-net of $\mathbb{S}^{k-1}$ with probability at least $1 - \delta$.*

*Proof.* We start by defining a partition $\pi : [n] \to [m]$ where $m = C(\log(1/\delta) + k\log(k))$ satisfying the following:

$$\forall j \in [m] : \sum_{i, \pi(i) = j} \frac{1}{\sigma_i^{4k-2}} \geqslant \frac{1}{2m} \sum_{i=1}^n \frac{1}{\sigma_i^{4k-2}}.$$

Such a partition may be found in linear time by Lemma C.7. For all $v \in \mathcal{G}, \ell \in [m]$, let $\widehat{\mathcal{D}}_{v,\ell}$ be the one-dimensional $k$-atomic mixture obtained by running Algorithm 3 on the points $\{\langle X_i, v\rangle\}_{i, \pi(i) = \ell}$. From Theorem 4.6, we get that:

$$\mathbb{P}\left\{ W_1(\mathcal{D}_v, \widehat{\mathcal{D}}_{v,\ell}) \leqslant Ck^2 \cdot \left( \sum_{i=1}^n \frac{m}{\sigma_i^{4k-2}} \right)^{-1/(4k-2)} \right\} \geqslant \frac{3}{4}.$$

Defining the random variables $Q_{v,\ell}$ as follows:

$$Q_{v,\ell} = \mathbf{1}\left\{ W_1(\mathcal{D}_v, \widehat{\mathcal{D}}_{v,\ell}) \leqslant Ck^2 \cdot \left( \sum_{i=1}^n \frac{m}{\sigma_i^{4k-2}} \right)^{-1/(4k-2)} \right\},$$

we get that $\mathbb{E}[Q_{v,\ell}] \geqslant 3/4$. Therefore, we get by an application of Hoeffding's inequality for a fixed $v \in \mathcal{G}$:

$$\mathbb{P}\left\{ \sum_{i=1}^m Q_{v,\ell} \leqslant 0.6m \right\} \leqslant \exp(-cm).$$

By a union bound, we get from our setting of $m$:

$$\mathbb{P}\left\{ \exists v \in \mathcal{G} : \sum_{i=1}^m Q_{v,\ell} \leqslant 0.6m \right\} \leqslant |\mathcal{G}| \exp(-cm) \leqslant \delta.$$

We now condition on the above event. At this point, we have $m$ estimates $\widehat{D}_{v,\ell}$ for each direction $v \in \mathcal{G}$. To obtain a single estimate for each direction, $v$, observe the following by the triangle inequality for any $j, j' \in [m]$ such that $Q_{v,j}, Q_{v,j'} = 1$:

$$W_1(\widehat{\mathcal{D}}_{v,j}, \widehat{\mathcal{D}}_{v,j'}) \leqslant W_1(\widehat{\mathcal{D}}_{v,j}, \mathcal{D}_v) + W_1(\mathcal{D}_v, \widehat{\mathcal{D}}_{v,j'}) \leqslant Ck^2 \cdot \left( \sum_{i=1}^n \frac{m}{\sigma_i^{4k-2}} \right)^{-1/(4k-2)}.$$

Now, as a consequence, the following set exists:

$$\mathcal{J} := \left\{ j \in [m] : \left| \left\{ \ell \in [m] : W_1(\widehat{\mathcal{D}}_{v,j}, \widehat{\mathcal{D}}_{v,\ell}) \leqslant Ck^2 \cdot \left( \sum_{i=1}^n \frac{m}{\sigma_i^{4k-2}} \right)^{-1/(4k-2)} \right\} \right| \geqslant 0.6m \right\}.$$

$\mathcal{J}$ is computable from the estimates $\widehat{\mathcal{D}}_{v,j}$ and we now show that *any* arbitrary element of $\mathcal{J}$ suffices. Let $j_v^* \in \mathcal{J}$ and notice by the pigeonhole principle and the definition of $\mathcal{J}$, that there exists $\ell \in [m]$ such that $Q_{v,\ell} = 1$ and furthermore,

$$W_1(\widehat{\mathcal{D}}_{v,j_v^*}, \widehat{\mathcal{D}}_{v,\ell}) \leqslant Ck^2 \cdot \left( \sum_{i=1}^n \frac{m}{\sigma_i^{4k-2}} \right)^{-1/(4k-2)} \implies W_1(\widehat{\mathcal{D}}_{v,j_v^*}, \mathcal{D}_v) \leqslant Ck^2 \cdot \left( \sum_{i=1}^n \frac{m}{\sigma_i^{4k-2}} \right)^{-1/(4k-2)}$$

by the triangle inequality and the definition of $Q_{v,\ell}$. The lemma follows from the estimates $\{\widehat{\mathcal{D}}_{v,j^*}\}$ for $v \in \mathcal{G}$. $\square$

Our algorithm uses a median-of-means based approach to recover the moments of the one-dimensional projections. Since the variances are non-uniform, we use a deterministic partitioning algorithm to partition points to ensure buckets of roughly equal size. The following lemma establishes guarantees on the sample partitioning algorithm.

**Lemma C.7.** *Given $k \in \mathbb{N}$ and $n$ numbers, $\alpha_1, \ldots, \alpha_n \geqslant 0$ such that $\max_i \alpha_i \leqslant \sum_{i=1}^n \alpha_i / 4k$, it is possible to find a partition $\pi : [n] \to [k]$ such that:*

$$\min_i \sum_{j:\pi(j)=i} \alpha_j \geqslant \frac{\sum_{i=1}^n \alpha_i}{2k}.$$

*Furthermore, the runtime of this procedure is $O(n \log(k))$.*

*Proof.* We will analyze the simple greedy procedure in Algorithm 6 where ties are broken arbitrarily. Define:

$$M_j^i := \sum_{\ell \leqslant i, \pi(\ell)=j} \alpha_\ell.$$

We will prove by induction on $i$ that:

$$\forall j, \ell \in [k] : |M_j^i - M_\ell^i| \leqslant \alpha_{\max}.$$

The case for $i = 1$ is trivially true. Suppose the condition is true for $i = m$. To extend to $m + 1$, let $j_M := \arg\max_i M_i^m$ and $j_m := \arg\min_i M_i^m$. Then, we must $\arg\max_j M_j^{m+1} \in \{j_M, j_m\}$. Note now that for all $\ell \neq j_M, j_m$:

$$|M_{j_M}^{m+1} - M_\ell^{m+1}| \leqslant \alpha_{\max}, \qquad |M_{j_m}^{m+1} - M_\ell^{m+1}| = |(M_\ell^m - M_{j_m}^m) - \alpha_{i+1}| \leqslant \alpha_{i+1} \leqslant \alpha_{\max}.$$

Finally, we have:

$$|M_{j_m}^{m+1} - M_{j_M}^{m+1}| = |(M_{j_M}^m - M_{j_m}^m) - \alpha_i| \leqslant \alpha_i \leqslant \alpha_{\max}$$

concluding the proof of the inductive claim and consequently, establishes the correctness of the algorithm. The runtime follows from the fact that the scores may be stored in a self-balancing binary tree.

---

**Algorithm 6** Partition Mass

---

**Input:** Weights $\alpha_1, \ldots, \alpha_n$
**Output:** Partition $\pi : [n] \to [k]$
 1: **for** $i = 1 \ldots n$ **do**
 2:     $\pi(i) = \arg\min_j \sum_{\ell < i, \pi(\ell)=j} \alpha_\ell$
 3: **end for**
 4: Return $\pi$

---

$\square$

In the next, we use the techniques of Doss et al. (2023) to improve the computational properties of the estimation algorithm. This part of the proof is essentially identical to Doss et al. (2023). However, we include minor changes which allow tightening the bounds. The next lemma shows that the set $\mathcal{C}$ defined in Algorithm 7, which defines the low-dimensional estimation algorithm, has a good solution.

**Lemma C.8.** *Let $d, k \in \mathbb{N}$ and let $\mathcal{D} = \sum_{i=1}^k w_i \delta_{\mu_i}$ be a $k$-atomic mixture over $\mathbb{R}^d$ with $\|\mu_i\| \leqslant R$ for all $i \in [k]$. Furthermore, for each $i \in [d]$, let $\widehat{\mathcal{D}}_{e_i} = \sum_{j=1}^k \widehat{w}_j^i \delta_{\mu_j^i}$ be $k$-atomic distributions over $\mathbb{R}$, each satisfying:*

$$\mathrm{W}_1(\mathcal{D}_{e_i}, \widehat{\mathcal{D}}_{e_i}) \leqslant \nu.$$

*Then, for any $(\nu/R)$-net over the probability simplex, $\mathcal{W}$, there exists a solution $\widehat{\mathcal{D}} = \sum_{i=1}^k \widehat{w}_i \delta_{\widehat{\mu}_i}$ such that:*

$$\mathrm{W}_1(\widehat{\mathcal{D}}, \mathcal{D}) \leqslant 2d\nu \text{ where } \widehat{w} \in \mathcal{W} \text{ and } \widehat{\mu}_i \in \{\widehat{\mu}_j^1\}_{j=1}^k \times \{\widehat{\mu}_j^2\}_{j=1}^k \times \cdots \times \{\widehat{\mu}_j^k\}_{j=1}^k.$$

*Proof.* We have as a consequence of the assumption of the lemma:

$$\forall j \in [d] : \sum_i w_i \min_{\ell \in [k]} |\langle \mu_i, e_j \rangle - \widehat{\mu}_\ell^j| \leqslant \nu.$$

---

**Algorithm 7** Low-dimensional Algorithm

---

**Input:** Data Points $X_1, \ldots, X_n$, variances $\sigma_1, \ldots, \sigma_n$ and failure probability $\delta$
**Output:** Low-dimensional mixture $\widehat{M} = \sum_{i=1}^{k} \widehat{w}_i \delta_{\widehat{\mu}_i}$

1: Let $\mathcal{G}$ be a $1/(32k^2\sqrt{k})$-net of $\mathbb{S}^{k-1}$ containing $e_1, \ldots, e_k$
2: For each $v \in \mathcal{G}$, let $\widehat{\mathcal{D}}_v$ be the output of Lemma C.6
3: Let for each $i \in [k]$, let $\widehat{\mathcal{D}}_{e_i} = \sum_{j=1}^{k} \widehat{w}_j^i \delta_{\widehat{\mu}_j^i}$

$$\mathcal{C} := \{\widehat{\mu}_i^1\}_{i=1}^{k} \times \{\widehat{\mu}_i^2\}_{i=1}^{k} \times \cdots \times \{\widehat{\mu}_i^k\}_{i=1}^{k}$$

4: Let $\mathcal{W}$ be a $(1/n^{1/(4k-2)})$-net over the probability simplex over $k$ elements
5: Define:

$$(\widehat{w}_i, \widehat{\mu}_i)_{i=1}^{n} := \underset{\widehat{w}' \in \mathcal{W}, \widehat{\mu}_i' \in \mathcal{C}}{\arg\min} \ \underset{v \in \mathcal{G}}{\arg\max} \ \mathrm{W}_1(\widehat{\mathcal{D}}_v, (\widehat{w}_i', \widehat{\mu}_i')_{i=1}^{k})$$

6: Return $\widehat{\mathcal{D}} = \sum_{i=1}^{k} \widehat{w}_i \delta_{\widehat{\mu}_i}$
Finally, let

---

Defining now:

$$\forall i \in [k] : \widehat{\mu}_i = (\widetilde{\mu}_i^1, \ldots, \widetilde{\mu}_i^d) \text{ where } \widehat{\mu}_\ell^j := \underset{\widetilde{\mu} \in \{\widehat{\mu}_m^j\}_{m=1}^k}{\arg\min} \ |\langle \mu_\ell, e_j \rangle - \widetilde{\mu}| \text{ for all } \ell \in [k], j \in [d].$$

We have for the mixture $\widetilde{\mathcal{D}} = \sum_{i=1}^{k} w_i \delta_{\widehat{\mu}_i}$:

$$\mathrm{W}_1(\mathcal{D}, \widetilde{\mathcal{D}}) \leqslant \sum_{i=1}^{k} w_i \|\mu_i - \widehat{\mu}_i\| \leqslant \sum_{i=1}^{k} w_i \|\mu_i - \widehat{\mu}_i\|_1 = \sum_{i=1}^{k} w_i \sum_{j=1}^{d} |\langle \mu_i, e_j \rangle - \widetilde{\mu}_i^j| = \sum_{j=1}^{d} \sum_{i=1}^{k} w_i |\langle \mu_i, e_j \rangle - \widetilde{\mu}_i^j| \leqslant d\nu.$$

Finally, let $\widehat{w} \in \mathcal{W}$ be such that $\|\widehat{w} - w\|_1 \leqslant \nu/R$. We have for the mixture $\widehat{\mathcal{D}} = \sum_{i=1}^{k} \widehat{w}_i \delta_{\widehat{\mu}_i}$:

$$\mathrm{W}_1(\mathcal{D}, \widehat{\mathcal{D}}) \leqslant \mathrm{W}_1(\mathcal{D}, \widetilde{\mathcal{D}}) + \mathrm{W}_1(\widetilde{\mathcal{D}}, \widehat{\mathcal{D}}) \leqslant d\nu + \sum_{i=1}^{k} |\widehat{w}_i - w_i| \max_{j,\ell \in [k]} \|\widehat{\mu}_j - \widehat{\mu}_\ell\| \leqslant d\nu + 2\sqrt{d}R \sum_{i=1}^{k} |\widehat{w}_i - w_i| \leqslant 2d\nu$$

concluding the proof of the lemma. $\qquad \square$

### C.3 PROOF OF THEOREM 4.2

The complete high-dimensional estimation algorithm is presented in Algorithm 8. The algorithm splits the points into two sets, and utilizes the first to compute a low-dimensional sub-space that (approximately) contains the means of the components of the mixture and then, runs the low-dimensional estimation algorithm Algorithm 7 on the second set. Before proceeding with the proof, we recall another lemma from Doss et al. (2023) which relates the loss in accuracy caused by the low-dimensional projection to the error in estimating the low-dimensional projection in Lemma C.4.

**Lemma C.9.** *Let $\mathcal{D} = \sum_{j=1}^{k} w_j \delta_{\mu_j}$ be a $k$-atomic distribution. Let $\Sigma = \mathbb{E}_{X \sim \mathcal{D}}[UU^\top] = \sum_{j=1}^{k} w_j \mu_j \mu_j^\top$ with eigenvalues $\lambda_1 \geqslant \cdots \geqslant \lambda_d$. Let $\Sigma'$ be a symmetric matrix and $\Pi_r'$ be the projection matrix onto the subspace spanned by the top $r$ eigenvectors of $\Sigma'$. Then,*

$$\mathrm{W}_1{}^2(\Gamma, \Gamma_{\Pi_r'}) \leqslant k \left( \lambda_{r+1} + 2\|\Sigma - \Sigma'\|_2 \right).$$

*Proof of Theorem 4.2.* The algorithm that achieves the guarantees of the theorem is defined in Algorithm 8. We first show that the partitioning step induces balanced partitions for the subspace computation and low-dimensional estimation steps.

**Claim C.10.** We have the following:

$$\sum_{X_i \in \boldsymbol{X}_1} \frac{1}{\sigma_i^4} \geqslant \frac{1}{4} \sum_{i=1}^{n} \frac{1}{\sigma_i^4}$$

---

**Algorithm 8** High-dimensional Gaussian Mixtures

---

**Input:** Point set $\boldsymbol{X} = \{X_1, \ldots, X_n\}$, Variances $\sigma_1, \ldots, \sigma_n$, Failure Probability $\delta$

**Output:** $k$-atomic mixture $\sum_{i=1}^{k} \widehat{w}_i \delta_{\widehat{\mu}_i}$

1: Partition $\boldsymbol{X}$ into $\boldsymbol{X}_1$ and $\boldsymbol{X}_2$ by allocating each $X_i$ to $\boldsymbol{X}_1$ or $\boldsymbol{X}_2$ uniformly at random
2: Define
$$\widehat{\Sigma} = \sum_{X_i \in \boldsymbol{X}_1} \alpha_i X_i X_i^\top - \bar{\sigma}^2 I \text{ where } \alpha_i := \frac{1/\sigma_i^4}{\sum_{X_i \in \boldsymbol{X}_1} 1/\sigma_i^4} \text{ and } \bar{\sigma}^2 = \sum_{X_i \in \boldsymbol{X}_1} \alpha_i \sigma_i^2.$$

3: Let $U$ be the top-$k$ singular subspace of $\widehat{\Sigma}$
4: Let $\widetilde{\boldsymbol{X}}_2 = \Pi_U(\boldsymbol{X}_2)$
5: Return $(\widehat{w}_i, \mu_i)_{i=1}^{k}$ as the output of Algorithm 7 on input $\widetilde{\boldsymbol{X}}_2$, weights $\{\sigma_i\}_{X_i \in \boldsymbol{X}_2}$, and failure probability $\delta/4$

---

$$\frac{c_2}{4(\log(1/\delta) + k\log(k)} \sum_{X_i \in \boldsymbol{X}_1} \frac{1}{\sigma_i^{4k-2}} \geqslant \frac{1}{4} \sum_{i=1}^{n} \frac{1}{\sigma_i^{4k-2}}$$

with probability at least $1 - \delta/4$.

*Proof.* For the first, observe that:

$$\sum_{X_i \in \boldsymbol{X}_1} \frac{1}{\sigma_i^4} = \sum_{i=1}^{n} W_i \text{ where } W_i := \frac{\mathbf{1}\{X_i \in \boldsymbol{X}_1\}}{\sigma_i^4}.$$

Now, we have by Hoeffding's inequality:

$$\mathbb{P}\left\{\sum_{i=1}^{n} W_i - \mathbb{E}\left[\sum_{i=1}^{n} W_i\right] \leqslant -t\right\} \leqslant \exp\left\{-\frac{2t^2}{\sum_{i=1}^{n} 1/\sigma_i^8}\right\} \leqslant \exp\left\{-2 \cdot \frac{t}{\max_i(1/\sigma_i^4)} \cdot \frac{t}{\sum_{i=1}^{n} 1/\sigma_i^4}\right\}$$

Now, setting:

$$t = \frac{1}{4} \sum_{i=1}^{n} \frac{1}{\sigma_i^4} \implies \frac{t}{\max_i(1/\sigma_i)^4} \cdot \frac{t}{\sum_{i=1}^{n} 1/\sigma_i^4} = \frac{1}{16} \cdot \frac{\sum_{i=1}^{n} 1/\sigma_i^4}{\max_i(1/\sigma_i^4)} \geqslant 4\log(1/\delta).$$

Observing that:

$$\mathbb{E}\left[\sum_{i=1}^{n} W_i\right] = \frac{1}{2} \sum_{i=1}^{n} \frac{1}{\sigma_i^4}$$

yields the first claim with probability at least $1 - \delta/8$. For the second, similarly define:

$$\sum_{X_i \in \boldsymbol{X}_2} \frac{1}{\sigma_i^{4k-2}} = \sum_{i=1}^{n} Q_i \text{ where } Q_i := \frac{\mathbf{1}\{X_i \in \boldsymbol{X}_2\}}{\sigma_i^{4k-2}}.$$

As before, we have by Hoeffding's inequality:

$$\mathbb{P}\left\{\sum_{i=1}^{n} Q_i - \mathbb{E}\left[\sum_{i=1}^{n} Q_i\right] \leqslant -t\right\} \leqslant \exp\left\{-\frac{2t^2}{\sum_{i=1}^{n} 1/\sigma_i^{8k-4}}\right\} \leqslant \exp\left\{-2 \cdot \frac{t}{\max_i(1/\sigma_i^{4k-2})} \cdot \frac{t}{\sum_{i=1}^{n} 1/\sigma_i^{4k-2}}\right\}$$

Now, setting:

$$t = \frac{1}{4} \sum_{i=1}^{n} \frac{1}{\sigma_i^{4k-2}} \implies \frac{t}{\max_i(1/\sigma_i)^{4k-2}} \cdot \frac{t}{\sum_{i=1}^{n} 1/\sigma_i^{4k-2}} = \frac{1}{16} \cdot \frac{\sum_{i=1}^{n} 1/\sigma_i^{4k-2}}{\max_i(1/\sigma_i^{4k-2})} \geqslant 4\log(1/\delta).$$

Observing,

$$\mathbb{E}\left[\sum_{i=1}^{n} Q_i\right] = \frac{1}{2} \sum_{i=1}^{n} \frac{1}{\sigma_i^{4k-2}}$$

yields the second claim with probability at least $1 - \delta/8$. A union bound establishes the claim. $\qquad \square$

We will now assume the event in the conclusion of Claim C.10 in the remainder of the proof. We have as a consequence of Claim C.10 and Lemma C.4:

$$\left\|\widehat{\Sigma} - \mathbb{E}_{X \sim \mathcal{D}}[XX^\top]\right\| \leqslant \max\left(\sqrt{(d + \log(1/\delta)) \sum_{X_i \in \boldsymbol{X}_1} \alpha_i^2 \sigma_i^4}, (d + \log(1/\delta)) \max_i \alpha_i \sigma_i^2, k\sqrt{\log(k/\delta) \sum_{X_i \in \boldsymbol{X}_1} \alpha_i^2}\right).$$

For the second term, we have for any $X_i \in \boldsymbol{X}_1$:

$$(d + \log(1/\delta))\alpha_i \sigma_i^2 \leqslant \sqrt{(d + \log(1/\delta))}\sqrt{(d + \log(1/\delta))\alpha_i^2 \sigma_i^4} \leqslant c\sqrt{\frac{d + \log(1/\delta)}{\sum_{X_i \in \boldsymbol{X}_1} 1/\sigma_i^4}}.$$

Hence, we get that the second is dominated by the first as:

$$\sum_{X_i \in \boldsymbol{X}_1} \alpha_i^2 \sigma_i^4 = \sum_{X_i \in \boldsymbol{X}_1} \frac{(1/\sigma_i)^4}{(\sum_{X_i \in \boldsymbol{X}_1} 1/\sigma_i^4)^2} = \frac{1}{\sum_{X_i \in \boldsymbol{X}_1} 1/\sigma_i^4}.$$

Finally, the third term is dominated by the second as $\sigma_i \geqslant 1$. To conclude, we get by Claim C.10 and the previous discussion that:

$$\left\|\widehat{\Sigma} - \mathbb{E}_{X \sim \mathcal{D}}[XX^\top]\right\| \leqslant \sqrt{\frac{(d + \log(1/\delta))}{\sum_{i=1}^n 1/\sigma_i^4}}.$$

Therefore, we have for the subspace $U$ in Algorithm 8 by Lemma C.9:

$$\mathrm{W}_1(\mathcal{D}, \mathcal{D}_U) \leqslant Ck\sqrt[4]{\frac{(d + \log(1/\delta))}{\sum_{i=1}^n 1/\sigma_i^4}}.$$

with probability at least $1 - \delta/4$. Next, since $\boldsymbol{X}_2$ are independent of $\boldsymbol{X}_1$ (and hence, the subspace $U$), we get by Claim C.10 and Lemmas C.6 and C.8 that the solution $\widehat{\mathcal{D}} = (\widehat{w}_i, \widehat{\mu}_i)_{i \in [k]}$ satisfies:

$$\mathrm{W}_1(\widehat{\mathcal{D}}, \mathcal{D}_U) \leqslant Ck^3 \left(\sum_{i=1}^k \frac{(k\log(k) + \log(1/\delta))}{\sigma_i^{4k-2}}\right)^{-1/(4k-2)}$$

with probability at least $1 - \delta/4$. By a union bound and the triangle inequality, we get:

$$\mathrm{W}_1(\widehat{\mathcal{D}}, \mathcal{D}) \leqslant \mathrm{W}_1(\widehat{\mathcal{D}}, \mathcal{D}_U) + \mathrm{W}_1(\mathcal{D}_U, \mathcal{D})$$

$$\leqslant C\left(k\left(\frac{(d + \log(1/\delta))}{\sum_{i=1}^n 1/\sigma_i^4}\right)^{1/4} + k^3\left(\sum_{i=1}^k \frac{(k\log(k) + \log(1/\delta))}{\sigma_i^{4k-2}}\right)^{-1/(4k-2)}\right)$$

with probability at least $1 - \delta$ establishing the correctness of Algorithm 8.

Finally, to bound the runtime, the computation of the top-$k$ right singular subspace can be computed in time $\widetilde{O}_k(nd)$ and projecting the points in $\boldsymbol{X}_2$ onto the low dimensional subspace is also computable in time $\widetilde{O}_k(nd)$. For the running of Algorithm 7, observe that the procedure in Lemma C.6 runs in time $\widetilde{O}_k(|\mathcal{G}|n) = \widetilde{O}_k(n)$. Finally, for the remainder of Algorithm 7, note that the number of candidate vectors in $\mathcal{C}$ is at most $k^k$ and the number of candidate weight vectors in $\mathcal{W}$ is at most $n$. Therefore, the total number of candidates is at most $\widetilde{O}_k(n)$. The time taken to check candidate solution is at most $\widetilde{O}_k(1)$ for each $v \in \mathcal{G}$ and since there are at most $\widetilde{O}_k(1)$ of these, the time to check a single candidate is $\widetilde{O}_k(1)$. Thus the total runtime for the Algorithm 7 is at most $\widetilde{O}_k(n)$ and as a consequence, the runtime of Algorithm 8 is bounded by $\widetilde{O}_k(nd)$. $\qquad\square$

## D LOWER BOUND: PROOF OF THEOREM 4.4

In this section, we prove the lower bound Theorem 4.4. We prove the lower bound in three cases which will imply the final bound. Firstly, we prove the $k$-dependent term from the lower bound in Appendix D.1 and show that it holds in the *one-dimensional* setting. Then, we establish a pair of bounds for high-dimensional setting, one for the dimension-dependent term in and another for the failure probability $\delta$, in Appendix D.2. We then combine these three results to obtain the final bound from Theorem 4.4. We will frequently make use of the following inequality which will be crucial in obtaining the high-probability lower bounds in the respective statements:

**Lemma D.1** ( Bretagnolle & Huber (1979); Tsybakov (2009); Canonne (2023)). *We have for any two distributions, $P$ and $Q$:*

$$\mathrm{TV}(P, Q) \leqslant 1 - \frac{1}{2} \exp(D_{\mathrm{KL}}(P\|Q)).$$

### D.1 ONE-DIMENSIONAL LOWER BOUND

Here, we establish a lower bound in *one* dimension for the estimation problem with $k$ components. We show that the $k$-dependent term in the lower bound is *unavoidable* even in the simpler one-dimensional setting. Our proof largely follows that in Wu & Yang (2020) but is specialized to the setting of heterogeneous variances. We start by recalling some technical results from Wu & Yang (2020).

**Lemma D.2.** *Let $\varepsilon \in [0, 1]$ and $l \in \mathbb{N}$. Suppose now that $\mathcal{D}$ and $\mathcal{D}'$ are centered distributions supported on $[-\varepsilon, \varepsilon]$ satisfying $\mathbf{m}_l(\mathcal{D}) = \mathbf{m}_l(\mathcal{D}')$. Then, we have the following:*

$$\chi^2(\mathcal{D} * \mathcal{N}(0, 1)\|\mathcal{D}' * \mathcal{N}(0, 1)) \leqslant O\left(\left(\frac{e\varepsilon^2}{l+1}\right)^{l+1}\right).$$

We also note the following simple corollary by the invariance of $f$-Divergences to re-scaling of the distributions.

**Corollary D.3.** *Let $\varepsilon \in [0, 1]$ and $l \in \mathbb{N}$. Suppose now that $\mathcal{D}$ and $\mathcal{D}'$ are centered distributions supported on $[-\varepsilon, \varepsilon]$ satisfying $\mathbf{m}_l(\mathcal{D}) = \mathbf{m}_l(\mathcal{D}')$. Then, we have the following for any $\sigma^2 \geqslant 1$:*

$$\chi^2(\mathcal{D} * \mathcal{N}(0, \sigma^2)\|\mathcal{D}' * \mathcal{N}(0, \sigma^2)) \leqslant O\left(\left(\frac{e\varepsilon^2}{(l+1)\sigma^2}\right)^{l+1}\right).$$

Lemma D.2 implies that two Dirac distributions supported on a small range and matching a large number of moments are challenging to distinguish from each other. The following result from Wu & Yang (2020) shows that when allowed a support size of $k$, there exist two distributions $\mathcal{D}$ and $\mathcal{D}'$ supported on $[-\varepsilon, \varepsilon]$ and more than $\varepsilon/k$ apart but nevertheless matching their first $4k - 2$ moments.

**Lemma D.4.** *For any $\varepsilon \geqslant 0$ and $k \in \mathbb{N}$, there exists two distributions $\mathcal{D}$ and $\mathcal{D}'$ supported on $[-\varepsilon, \varepsilon]$ satisfying:*

$$\mathrm{W}_1(\mathcal{D}, \mathcal{D}') \geqslant \Omega\left(\frac{\varepsilon}{k}\right) \quad and \quad \mathbf{m}_{2k-2}(\mathcal{D}) = \mathbf{m}_{2k-2}(\mathcal{D}').$$

*Furthermore, $\mathcal{D}$ and $\mathcal{D}'$ are supported on at most $k$ elements.*

Next, we prove the main one-dimensional lower bound.

**Theorem D.5.** *There exists an absolute constants $C, c > 0$ such that the following holds. Let $n, k \in \mathbb{N}$, $\delta \in (0, 1/4]$, and $\sigma_i \geqslant 1$ for $i \in [n]$ be such that:*

$$\sum_{i=1}^{n} \frac{1}{\sigma_i^{4k-2}} \geqslant (C\sqrt{k})^{4k-2} \log(1/\delta).$$

*Then, there exist two distributions, $\mathcal{D}_1$ and $\mathcal{D}_2$, each supported on at most $k$-points in $[-1, 1]$, such that the following holds for any estimator $\widehat{\theta}$:*

$$\max_{\mathcal{D} \in \{\mathcal{D}_1, \mathcal{D}_2\}} \mathbb{P}_{\boldsymbol{X}} \left\{ \mathrm{W}_1(\mathcal{D}, \widehat{\theta}(\boldsymbol{X})) \geqslant \frac{c}{k} \left(\frac{1}{\log(1/\delta)} \sum_{i=1}^{n} \left(\frac{1}{\sigma_i^{4k-2}}\right)\right)^{-1/(4k-2)} \right\} \geqslant \delta \text{ where } \boldsymbol{X} \sim \prod_{i=1}^{n} \mathcal{D} * \mathcal{N}(0, \sigma_i^2).$$

*Proof.* Let $\varepsilon \in [0, 1]$ be a parameter to be chosen later and $\mathcal{D}$ and $\mathcal{D}'$ be two distributions whose existence is guaranteed by Lemma D.4. Then, observe that we have the following for all $i \in [n]$:

$$\chi^2(\mathcal{D} * \mathcal{N}(0, \sigma_i^2)\|\mathcal{D}' * \mathcal{N}(0, \sigma_i^2)) \leqslant O\left(\left(\frac{e\varepsilon^2}{(2k-1)\sigma_i^2}\right)^{2k-1}\right).$$

Recall the tensorization properties of the $\chi_2$ divergence:

$$\chi_2\left(\prod_{i=1}^{n} \mathcal{D} * \mathcal{N}(0, \sigma_i^2)\|\prod_{i=1}^{n} \mathcal{D}' * \mathcal{N}(0, \sigma_i^2)\right) + 1 = \prod_{i=1}^{n} \left(\chi_2\left(\mathcal{D} * \mathcal{N}(0, \sigma_i^2)\|\mathcal{D}' * \mathcal{N}(0, \sigma_i^2)\right) + 1\right).$$

As a consequence, we get that:

$$\chi_2\left(\prod_{i=1}^{n}\mathcal{D}*\mathcal{N}(0,\sigma_i^2)\|\prod_{i=1}^{n}\mathcal{D}'*\mathcal{N}(0,\sigma_i^2)\right)\leqslant C\sum_{i=1}^{n}\left(\frac{e\varepsilon^2}{(2k-1)\sigma_i^2}\right)^{2k-1}\quad\text{when}\quad\sum_{i=1}^{n}\left(\frac{e\varepsilon^2}{(2k-1)\sigma_i^2}\right)^{2k-1}\leqslant c$$

for some absolute constants $C, c > 0$. Finally, picking $\varepsilon$ as:

$$\varepsilon = c\left(\frac{1}{\log(1/\delta)}\sum_{i=1}^{n}\left(\frac{e}{(2k-1)\sigma_i^2}\right)^{2k-1}\right)^{-1/(4k-2)}$$

for $c > 0$ yields:

$$\chi_2\left(\prod_{i=1}^{n}\mathcal{D}*\mathcal{N}(0,\sigma_i^2)\|\prod_{i=1}^{n}\mathcal{D}'*\mathcal{N}(0,\sigma_i^2)\right)\leqslant\frac{\log(1/\delta)}{8}.$$

As a consequence, we get:

$$\gamma := D_{\mathrm{KL}}\left(\prod_{i=1}^{n}\mathcal{D}*\mathcal{N}(0,\sigma_i^2)\|\prod_{i=1}^{n}\mathcal{D}'*\mathcal{N}(0,\sigma_i^2)\right)\leqslant\frac{\log(1/\delta)}{8}.$$

By the BH-inequality (Lemma D.1), we now get:

$$\mathrm{TV}\left(\prod_{i=1}^{n}\mathcal{D}*\mathcal{N}(0,\sigma_i^2)\|\prod_{i=1}^{n}\mathcal{D}'*\mathcal{N}(0,\sigma_i^2)\right)\leqslant 1-\frac{1}{2}\exp\left(-\gamma\right)\leqslant 1-\delta.$$

An application of Le Cam's (Wainwright, 2019, Lemma 15.9) method now yields the result. $\qquad\square$

## D.2 High-dimensional Lower Bounds

Here, we adapt the techniques of Wu & Zhou (2021) to establish lower bounds in the setting of heterogenous variances. The dimension-dependent lower bound is established for the simple setting with two equally weighted components with means distributed symmetrically across the origin. Hence, the set of candidates mixtures $\mathcal{M}_{\mathrm{Dirac}}$ parameterized by $\theta \in \mathbb{R}^d$ and defined below:

$$\mathcal{M}_{\mathrm{Dirac}} = \frac{1}{2}\delta_\theta + \frac{1}{2}\delta_{-\theta}.$$

In the rest of this section, we will use $\mathcal{D}_\theta$ to refer to a member of this family. We now reproduce a technical result from Wu & Zhou (2021) which bounds the KL-Divergence between two different Gaussian mixtures from this family in terms of the distance between their respective location parameters.

**Lemma D.6.** *Then there exists a universal constant $C$ such that the following holds. For any $s \in [0,1]$ and $u, v$ such that $\|u\| = \|v\| = 1$, we have:*

$$D_{\mathrm{KL}}(\mathcal{D}_{su}*\mathcal{N}(0,I)\|\mathcal{D}_{sv}*\mathcal{N}(0,I))\leqslant C\|u-v\|^2 s^4.$$

Furthermore, we have by the scale-invariance of the KL-divergence, the following corollary.

**Corollary D.7.** *Assume the setting of Lemma D.6. Then, for any $\sigma \geqslant 1$, we have:*

$$D_{\mathrm{KL}}(\mathcal{D}_{su}*\mathcal{N}(0,\sigma^2 I)\|\mathcal{D}_{sv}*\mathcal{N}(0,\sigma^2 I))\leqslant C\|u-v\|^2\left(\frac{s}{\sigma}\right)^4.$$

We now present the main lower bound of the section:

**Theorem D.8.** *There exist absolute constants $C, c > 0$ such that the following holds. Let $d \geqslant 3$, $n \in \mathbb{N}$, and $\{\sigma_i\}_{i=1}^{n}$ be such that $\sigma_i \geqslant 1$ for all $i \in [n]$ and:*

$$\sum_{i=1}^{n}\frac{1}{\sigma_i^4}\geqslant Cd.$$

*Then, we have for any estimator $\widehat{T}$:*

$$\max_{\theta\in\mathbb{R}^d}\mathbb{P}_{\boldsymbol{X}}\left\{\mathrm{W}_1(\widehat{T}(\boldsymbol{X}),\mathcal{D}_\theta)\geqslant c\left(\frac{d}{\sum_{i=1}^{n}1/\sigma_i^4}\right)^{1/4}\right\}\geqslant\frac{1}{2}$$

*where $\boldsymbol{X} = \{X_i\}_{i=1}^{n}$ are independent random vectors with $X_i \sim \mathcal{D}_\theta * \mathcal{N}(0, \sigma_i^2 I)$.*

*Proof.* The proof will use the standard testing to estimation framework for proving minimax lower bounds. First, let $\mathcal{G}$ denote a $1/4$-packing of the unit sphere such that $|\mathcal{G}| \geqslant 4^{d-1}$. Now, we have from Corollary D.7, for any $u, v \in \mathcal{G}$ and $\sigma \geqslant 1$ and $s \in [0, 1]$:

$$D_{\mathrm{KL}}(\mathcal{D}_{su} * \mathcal{N}(0, \sigma^2 I) \| \mathcal{D}_{sv} * \mathcal{N}(0, \sigma^2 I)) \leqslant C \|u - v\|^2 \left(\frac{s}{\sigma}\right)^4 \leqslant 4C \left(\frac{s}{\sigma}\right)^4.$$

Hence, we get by the tensorization of the KL-divergence:

$$D_{\mathrm{KL}}\left(\prod_{i=1}^n \mathcal{D}_{su} * \mathcal{N}(0, \sigma_i^2 I) \middle\| \prod_{i=1}^n \mathcal{D}_{sv} * \mathcal{N}(0, \sigma_i^2 I)\right)$$
$$= \sum_{i=1}^n D_{\mathrm{KL}}\left(\mathcal{D}_{su} * \mathcal{N}(0, \sigma_i^2 I) \middle\| \mathcal{D}_{sv} * \mathcal{N}(0, \sigma_i^2 I)\right) \leqslant C s^4 \sum_{i=1}^n \frac{1}{\sigma_i^4}.$$

Now, picking

$$s = c \left(\frac{d}{\sum_{i=1}^n 1/\sigma_i^4}\right)^{1/4},$$

we get that:

$$D_{\mathrm{KL}}\left(\prod_{i=1}^n \mathcal{D}_{su} * \mathcal{N}(0, \sigma_i^2 I) \middle\| \prod_{i=1}^n \mathcal{D}_{sv} * \mathcal{N}(0, \sigma_i^2 I)\right) \leqslant \frac{d}{16}.$$

An application of Fano's inequality (Wainwright, 2019, Proposition 15.12) for the set of candidates $\{su\}_{u \in \mathcal{G}}$ and noting that $|\mathcal{G}| \geqslant 4^{d-1}$ yields the result from the observation that for any $u, v \in \mathcal{G}$, $\mathrm{W}_1(\mathcal{D}_{su}, \mathcal{D}_{sv}) \geqslant s/4$. $\square$

We now establish a separate high-probability version of the bound.

**Theorem D.9.** *There exist absolute constants $C, c > 0$ such that the following holds. Let $d \geqslant 2$, $n \in \mathbb{N}$, $\delta \in (0, 1/4]$, and $\{\sigma_i\}_{i=1}^n$ be such that $\sigma_i \geqslant 1$ for all $i \in [n]$ and:*

$$\sum_{i=1}^n \frac{1}{\sigma_i^4} \geqslant C \log(1/\delta).$$

*Then, we have for any estimator $\widehat{T}$:*

$$\max_{\theta \in \mathbb{R}^d} \mathbb{P}_{\boldsymbol{X}}\left\{\mathrm{W}_1(\widehat{T}(\boldsymbol{X}), \mathcal{D}_\theta) \geqslant c \left(\frac{\log(1/\delta)}{\sum_{i=1}^n 1/\sigma_i^4}\right)^{1/4}\right\} \geqslant \frac{1}{2}$$

*where $\boldsymbol{X} = \{X_i\}_{i=1}^n$ are independent random vectors with $X_i \sim \mathcal{D}_\theta * \mathcal{N}(0, \sigma_i^2 I)$.*

*Proof.* Let $u = e_1$ and $v = e_2$. Then, we have by Corollary D.7 for any $\sigma \geqslant 1$ and $s \in [0, 1]$:

$$D_{\mathrm{KL}}(\mathcal{D}_{su} * \mathcal{N}(0, \sigma^2 I) \| \mathcal{D}_{sv} * \mathcal{N}(0, \sigma^2 I)) \leqslant 2C \left(\frac{s}{\sigma}\right)^4.$$

We now have by the tensorization of the KL-divergence:

$$D_{\mathrm{KL}}\left(\prod_{i=1}^n \mathcal{D}_{su} * \mathcal{N}(0, \sigma_i^2 I) \middle\| \prod_{i=1}^n \mathcal{D}_{sv} * \mathcal{N}(0, \sigma_i^2 I)\right)$$
$$= \sum_{i=1}^n D_{\mathrm{KL}}\left(\mathcal{D}_{su} * \mathcal{N}(0, \sigma_i^2 I) \middle\| \mathcal{D}_{sv} * \mathcal{N}(0, \sigma_i^2 I)\right) \leqslant C s^4 \sum_{i=1}^n \frac{1}{\sigma_i^4}.$$

Now, setting:

$$s = c \left(\frac{\log(1/\delta)}{\sum_{i=1}^n 1/\sigma_i^4}\right)^{1/4},$$

we get:

$$D_{\mathrm{KL}}\left(\prod_{i=1}^n \mathcal{D}_{su} * \mathcal{N}(0, \sigma_i^2 I) \middle\| \prod_{i=1}^n \mathcal{D}_{sv} * \mathcal{N}(0, \sigma_i^2 I)\right) \leqslant \frac{\log(1/\delta)}{8}.$$

Again, we get by the BH-inequality (Lemma D.1):

$$\mathrm{TV}\left(\prod_{i=1}^{n}\mathcal{D}_{su}*\mathcal{N}(0,\sigma_i^2 I),\prod_{i=1}^{n}\mathcal{D}_{sv}*\mathcal{N}(0,\sigma_i^2 I)\right)\leqslant 1-\delta.$$

An application of Le Cam's method (Wainwright, 2019, Lemma 15.9) concludes the proof of the result. □

Finally, we combine the three bounds to prove the complete lower bound from Theorem 4.4.

*Proof of Theorem 4.4.* The proof follows from Theorems D.5, D.8 and D.9 by choosing the largest out of the three lower bounds in the conclusions of the results. □

# E  ADDITIONAL RESULTS

## E.1  PUSHING PERFORMANCE TO ITS LIMIT

In all our previous experiments, we used the EDM (Karras et al., 2022) hyperparameters that were tuned for clean data training. Here, we ablate different parameters that can further increase the performance of training with data mixtures. The first observation is that data corruption increases the time for convergence. In Figure 4 we plot the FID as a function of training time for a model trained with data mixtures. As shown, the FID keeps dropping as we train, indicating that the default parameters might lead to undertrained models when there is training data corruption. Hence, we take the ImageNet training run with $90\%$ noisy data and $10\%$ clean data, which has reported performance $1.68$ in Table 3 and we train the model for 500K more steps. This alone decreases the FID from $1.68$ to $1.63$.

We then argue that for times $t \leqslant t_n$ we should expect sub-optimal estimates since only a few (clean) samples were used. We attempt to balance the higher learning error with reduced discretization error by increasing the number of sampling steps spent on these noise levels during sampling. This further improves the performance from $1.63$ to $1.60$.

Next, we use consistency loss to improve the learning for $t < t_n$, that was previously trained only with the clean points. We note that we only use consistency loss for this final part of the 500k training steps. Consistency fine-tuning improves the FID to $1.58$. Finally, we use weight decay as a regularization technique to mitigate instabilities caused by the increased training variance. This further reduces the FID to $1.55$. It is possible that more optimizations could further decrease the FID but at this point we are already close to the optimal $1.41$ obtained using 100% clean samples.

| Configuration ($\sigma = 0.2$) $p = 10\%$ clean and $90\%$ noisy | ImageNet FID |
|---|---|
| A Baseline Parameters | 1.68 |
| B + More training | 1.63 |
| C + Sampling Noise Scheduling | 1.60 |
| D + Consistency Loss | 1.58 |
| E + Weight Decay | 1.55 |
| Clean Data Performance | 1.41 |

Figure 2: Evaluation of training and sampling improvements for models trained with noisy data.

In what follows, we present some additional results that did not fit in the main paper.

CIFAR          CelebA          ImageNet

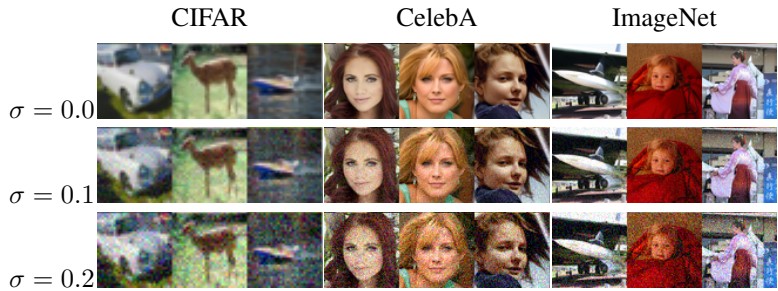

Figure 3: Dataset images with varying noise levels ($\sigma$).

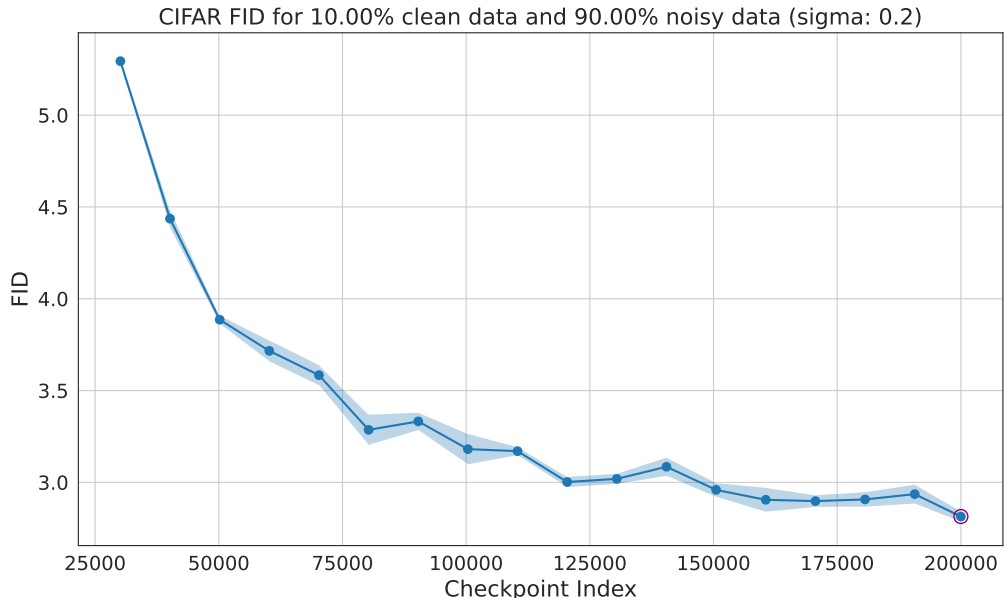

Figure 4: FID as a function of training steps for a model trained with a mix of clean and noisy data. FID continues to go down as we train more and more, indicating that the model at 200K iterations is still undertrained.

## F    EXPERIMENTAL DETAILS

### F.1    TRAINING

We train our CIFAR-10 models for 200,000 steps, our CelebA models for 100,000 steps and our ImageNet models for $2,500,000$ steps. All of our images are normalized in $[-1, 1]$ prior to adding additive Gaussian Noise. We use the training hyperparameters from the EDM codebase (Karras et al., 2022). For our experiments with consistency, we use 8 steps for the reverse sampling and 32 samples to estimate the expectations. We use a fixed coefficient to weight the consistency loss that is chosen as a hyperparameter from the set of $\{0.1, 1.0, 10.0\}$ to maximize performance. Upon acceptance of this work, we will provide all the code and checkpoints to accelerate research in this area. Consistency leads to a $2\times$ increase in hardware requirements (memory footprint) and $2\times$ (ImageNet) to $3\times$ (CIFAR, CelebA) slowdown.

### F.2    EVALUATION

We generate 50,000 images to compute FID. Each FID number reported in this paper is the average of three independent FID computations that correspond to the seeds: 0-49,999, 50,000-99,999, 100,000-149999.

