# OpenReview forum: "How Much is a  Noisy Image Worth? Data Scaling Laws for Ambient Diffusion."
_ICLR.cc/2025/Conference — ICLR 2025 Poster_

### Official Review · Reviewer_kj8h · 2024-10-27

**Soundness:** 3
**Presentation:** 4
**Contribution:** 3
**Rating:** 6
**Confidence:** 3

**Summary:**

This paper tackles a critical issue in training diffusion models with corrupted data when clean data is scarce or costly. The authors explore the effectiveness of a combination of a small clean dataset and a large noisy dataset, demonstrating that this approach can yield performance close to models trained solely on clean data. Theoretical analysis based on Gaussian Mixture Models (GMMs) with heterogeneous variances and experimental validation across multiple datasets support these claims. Additionally, the paper proposes a budget allocation mechanism for more efficient dataset curation by balancing the use of noisy and clean data.

**Strengths:**

* The paper is well-written and logically structured, making the content easy to follow.
* The theoretical analysis is rigorous and provides good intuition behind the proposed methods. The experiments are thorough, involving the training of more than 80 models across various datasets and noise levels.
* The paper’s conclusions offer significant practical value for dataset curation and budget allocation strategies in real-world applications, where collecting clean data can be expensive.

**Weaknesses:**

* The analysis is limited to Gaussian noise corruption and discrete distributions with finite support. Expanding this framework to continuous distributions and non-Gaussian noise would increase its impact.
* The proposed method assumes that the noise level is known in advance, which may not be the case in real-world scenarios. Incorporating techniques and analysis to handle unknown noise levels would make the approach more practical.

**Questions:**

It remains unclear why an exponentially larger amount of corrupted data is necessary to compensate for the lack of clean data. Can the authors provide more clarification for this point?

---

> ### Author Response · Authors · 2024-11-20
> **Response to Reviewer kj8h**
>
> We thank the Reviewer for their time and high-quality review. We are very glad that the Reviewer appreciated our work. In what follows, we provide clarifications to the Reviewer's feedback.
>
>
> ---
>
> **Feedback**: The analysis is limited to Gaussian noise corruption and discrete distributions with finite support [...]
>
> **Response**: This is true and we acknowledge this in our Limitations section (Section 6).
>
> This work shows for the first time that models trained with noisy data can achieve performance comparable to models trained with clean data. The next step is to generalize this to arbitrary corruptions. We are actively working on it and we have preliminary positive results in this direction. That said, including them in this submission is not feasible due to space, time and scope constraints.
>
> We also note that the mixture-of-Gaussians setting is a simple model which avoids the statistical intractability of high-dimensional non-parametric estimation. It is known that for the general class of continuous distributions a rate of n^{-1/d} is inevitable due to known minimax lower bounds, thus suffering from a curse of dimensionality. Thus, the Gaussian mixture model setting provides a statistically tractable alternative with an interpretable notion of distributional complexity. This model also helps elucidate the value of clean and noisy samples in distribution learning.
>
> ---
>
> **Feedback**: The proposed method assumes that the noise level is known in advance [...]
>
> **Response**: That’s a great point. We actually **do not need to know** the value of std since **this can be estimated**. Specifically, one can compute the difference of the variance between the clean and bad sets and this will give an estimation of the noise level of the noisy set. We added this discussion in the revised manuscript and we thank the Reviewer for helping us improve our work.
>
> ---
>
> **Feedback**: It remains unclear why an exponentially larger amount of corrupted data is necessary to compensate for the lack of clean data [...]
>
> **Response**:  Great question!
>
> Each sample in the dataset carries some information about the distribution of interest. The more noisy it is, the less information it carries.
>
> In our theoretical results, we define the notion of the effective number of samples. This quantity captures how much actual information we have in our training set, accounting for the different noise levels of the training samples. The contribution of each noisy sample to this quantity is exponentially smaller compared to the contribution of a clean sample.

---

> > ### Comment · Reviewer_kj8h · 2024-11-24
> > **reviewer answer**
> >
> > I thank the reponse of the authors and choose to keep my score. Moreover, as the authors said the variance can be estimated, it might be more reasonable to consider the sample complexity for getting a good estimate itself into the analysis.

---

> > > ### Author Response · Authors · 2024-11-24
> > > **Thank you and further clarification**
> > >
> > > Thank you for acknowledging our rebuttal. Since the noise is Gaussian, the sample complexity for variance estimation is well understood. We will include this analysis in our camera-ready version, together with empirical estimation results in the considered datasets.
> > >
> > > Please let us know if there is anything we could do to make you consider increasing further your rating. In any case, we sincerely thank you for helping us strengthen our submission.

---

### Official Review · Reviewer_vSSn · 2024-11-01

**Soundness:** 3
**Presentation:** 3
**Contribution:** 4
**Rating:** 8
**Confidence:** 3

**Summary:**

First, the paper empirically shows that training on entire noisy data will lead to bad performance, but as long as a few clean data (~10%) are available, models trained by the proposed algorithm are comparable to models trained on clean data. Second, the paper derives minimax optimal estimation bounds to explain the phenomenon. Third, the paper pinpoints the price of noisy images for different datasets and noise levels, which has practical implications.

**Strengths:**

1. The paper is written clearly.
2. In practice, we have small sets of clean data and large sets of noisy data, so the paper considers an important setting.
3. The experiments on different realistic image datasets are sufficient to support the claim in the paper.
4. The established estimation bounds are minimax optimal and discussed clearly.
5. The price of noisy images is interesting.

**Weaknesses:**

1. The obtained $\hat{D}$ is the ERM result for the Wasserstein distance loss, which is mismatched with the loss of diffusion models. Thus, there exists a gap between theory and practice.
2. The proposed algorithm and proof technique are mainly based on existing works, which may decrease the novelty of this paper.

**Questions:**

1. How can we know the value $\sigma_{t_n}$ in the inputs of Algorithm 1?

---

> ### Author Response · Authors · 2024-11-20
> **Response to Reviewer vSSn**
>
> We thank the Reviewer for their time and their high-quality review. We are very glad that the Reviewer appreciated our work. In what follows, we provide clarifications to the Reviewer's questions.
>
> ---
>
> **Feedback**: The obtained D^  is the ERM result for the Wasserstein distance loss [...] not the diffusion loss [...]
>
> **Response**: Great question. The goal of the training of a diffusion model is to learn the underlying distribution of the data. The training loss is a proxy of how well the model is doing in this direction. In fact, there are formal connections between the diffusion model training objective and the minimization of Wasserstein distance, as shown in the paper: [Score-based Generative Modeling Secretly Minimizes the Wasserstein Distance](https://arxiv.org/abs/2212.06359). The minimax optimality of our results thus provides a tight characterization of the statistical values of samples of different noise levels in learning the underlying clean distribution. Hence, these bounds also apply to the training of diffusion models which aim to learn the underlying distribution, formally connecting the two settings.
>
> ---
>
> **Feedback**: The proposed algorithm and proof technique are mainly based on existing works, which may decrease the novelty of this paper.
>
> **Response**: The main contribution of our theoretical results is a formalization of the value of clean and noisy samples in learning the underlying distribution, a factor not present in prior work. This requires a modification of the algorithm to take into account these different levels. Furthermore, these modifications crucially depend on the particular step of the algorithm. Consequently, the dimensionality reduction and low-dimensional estimation steps require two different weighing schemes showcasing the different utilities of these different types of samples for these different sub-steps of the learning process. The optimality of these bounds furthermore establish that these trade-offs are inherent and not the result of the particular algorithm developed in our work. Finally, we also substantially improve on the analysis in previous work even for the setting with no variances in sample quality as our results yield optimal high-probability guarantees not present in previous work.
>
> ---
>
> **Feedback**: How can we know the value of sigma_n?
>
> **Response**: That’s a great question. We actually do not need to know the actual value of std since this can be estimated. Specifically, one can compute the difference of the variance between the clean and bad sets and this will give an estimation of the noise level of the noisy set. We added this discussion in the revised manuscript and we thank the Reviewer for helping us improve our work.

---

> > ### Comment · Reviewer_vSSn · 2024-11-25
> >
> > Thank the authors for the detailed reply. Since some of my concerns have been addressed, I will maintain my positive score.

---

### Official Review · Reviewer_zaEs · 2024-11-02

**Soundness:** 4
**Presentation:** 3
**Contribution:** 3
**Rating:** 8
**Confidence:** 3

**Summary:**

Quick summary: This paper studies the performance of the diffusion model in the ambient diffusion scenario. Previous work shows that the diffusion model trained by noisy data is inferior to the model trained on clean data, and the consistent diffusion model requires the noisy samples to be infinity from the asymptotic view. This paper proposes to use the mixture of clean data (high-frequency feature) and corrupted data (low-frequency feature) to assist the ambient diffusion model to obtain a very close performance to the state-of-the-art and provide the theoretic analysis to support the experiment and evaluate the corresponding value of noise image using the inspiring sampling bounds from the theoretical results.
Quality: The paper is well-written and well-motivated, and the proposed method is good with a theoretical foundation albeit a bit simple.
Originality: The theoretical analysis of the two sampling bounds is inspiring with practical meaning.
Significance: This is an important research direction because noise data is common in the real world.
Pros: * Important problem * theoretical foundation * nice results and sufficient experiments
Cons: * The solution is a bit simple and kind of engineering consideration.
Summary: This is a nice paper that gives a simple solution to the ambient diffusion models with theoretic analysis and while not groundbreaking, certainly merits a publication.

**Strengths:**

The theoretical analysis of the two sampling bounds is inspiring with practical meaning.

**Weaknesses:**

The solution is a bit simple and kind of engineering consideration.

**Questions:**

What's the exact number of samples or dimensions in your settings in line 93, page 2?

---

> ### Author Response · Authors · 2024-11-20
> **Response to Reviewer zaEs**
>
> We thank the Reviewer for their time and for their input. We are very glad that the Reviewer appreciated our work!
>
> ---
>
> **Feedback**: The solution is a bit simple.
>
> **Response**: We agree with the Reviewer regarding the simplicity of the solution, but we view this positively because it might allow wider adoption. The consistency loss of Daras et. al, albeit necessary when all of the data is corrupted, adds a significant computational overhead and training instability. In this paper we show that as long as a few clean data points are available, the solution can be drastically simplified and the results can be significantly improved.
>
> ---
>
> **Question**:  What's the exact number of samples or dimensions in your settings in line 93, page 2?
>
> **Response**: Our theoretical bounds are applicable for any number of samples, dimensions or noise levels. To match our experimental settings, the dimension is necessarily large while the number of classes is quite small. The number of samples, however, is challenging to state since the statistical value of each sample depends on its noise level and which step of the learning process it is used in. For the observations in line 93, the number of noisy samples is larger than the dimension. Furthermore, the scale of the added noise for the noisy samples also grows polynomially with dimension and the number of clean samples only grows with the number of classes. In the regime of large enough dataset, the second term of the Theorem 4.2 dominates and that characterizes the effective number of samples (accounting for their different utilities). Roughly speaking, the effective number of samples should grow as fast as k^k to achieve low estimation error in the theoretical result. Notice that we our sample dependency here does not grow exponentially with d, so in the studied model we escape the curse of dimensionality because of the structure of the studied distribution).

---

> > ### Comment · Reviewer_zaEs · 2024-11-26
> >
> > Apologies for the late response. Thanks for the response from the author.
> >
> > For the first point, I am glad to see how the engineering-focused solution can be generalized with deep theoretical analysis.
> >
> > For the second point, thanks for the detailed explanation.
> >
> > After reading all the comments, I plan to keep my score.

---

### Official Review · Reviewer_brqJ · 2024-11-04

**Soundness:** 3
**Presentation:** 3
**Contribution:** 3
**Rating:** 6
**Confidence:** 2

**Summary:**

This paper studies data scaling laws for ambient diffusion. The authors find that training on a small set of clean data and a large set of noisy data significantly outperforms training solely on either one of these sets. The paper provides both theoretical analysis and empirical results to verify their findings.

**Strengths:**

1. The studied problem is interesting and important.
2. The authors provide theoretical results to support their findings.

**Weaknesses:**

1. I'm a bit confused about the relationship between thm 4.2 and alg 1. The thm 4.2 states that there exists a procedure which can return a good estimate of D. Is Algorithm 1 the procedure referenced in this theorem?
2. Alg 1 need to know whether each data is clean or noisy. Is it possible to generalize the proposed algorithm to handle cases where clean and noisy data are mixed?
3. In the experiments, The paper only presents the empirical results of the proposed method and does not compare it with previous work. (Daras et al., 2024; 2023b).

**Questions:**

see weakness

---

> ### Author Response · Authors · 2024-11-20
> **Response to Reviewer brqJ**
>
> We thank the Reviewer for their time and input. In what follows, we do our best to reply to the Reviewer's questions.
>
> ---
>
> **Feedback**: I'm a bit confused about the relationship between thm 4.2 and alg 1. The thm 4.2 states that there exists a procedure which can return a good estimate of D. Is Algorithm 1 the procedure referenced in this theorem?
>
> **Response**: We thank the Reviewer for their question. Algorithm 3 is the procedure referenced in Theorem 4.2. We updated the submission and made this explicit.
>
> Our developed method (Algorithm 1) can be used to learn arbitrary distributions given noisy samples from this distribution at different levels of noise. Our experimental results suggest that a small number of clean samples is enough in practice, as long as a lot of noisy samples are available.
>
> This motivated us to look at a well-studied distribution in the diffusion theory world, the Gaussian Mixture Model, and understand the sample complexity bounds. Our theory develops matching lower and upper bounds for the estimation error given a finite set of noisy samples at different levels of noise. Algorithm 3 is the algorithm used to produce our upper bound on the estimation error. It is highly non-trivial to characterize if the algorithm we run in practice (Algorithm 1) achieves this optimal estimation error. The value of the developed theoretical results is that the expected behavior of the optimal algorithm (at least for GMMs) matches the experimental observations about the value of noisy vs clean data.
>
> Once again, we thank the Reviewer for raising this point and we clarified this in the paper.
>
> ---
>
> **Feedback**: Is it possible to generalize the proposed algorithm to handle cases where clean and noisy data are mixed?
>
> **Response**: Great question! We note without any information on the relative composition of clean and noisy samples, this problem is **information-theoretically impossible**: given a set of samples, it is not possible to determine whether they were generated entirely from the clean distribution or from a mixture of the clean and noisy distributions. On the other hand, if the composition is known in advance (say 10% clean and 90% noisy), it is possible to information-theoretically determine the clean distribution from the samples through the Fourier transform. However, it is unclear how to incorporate this observation towards improving Algorithm 1. Addressing this limitation is an exciting direction for future work!
>
> ---
>
> **Feedback**: The paper only presents the empirical results of the proposed method and does not compare it with previous work. (Daras et al., 2024; 2023b).
>
>
> **Response**: The Reviewer has a **misconception** here. The referenced prior work assumes that 100% of the data is corrupted. We include the comparison with this setting and we show that having 100% corrupted data leads to a significant performance deterioration. Our method is a proper generalization of Daras et. al. 2024 to account for settings where there are a few uncorrupted data points. We show that the addition of a few clean data points significantly increases the performance.
>
>
> ---
>
> We hope that we addressed some of the concerns of the Reviewer and that the Reviewer will consider upgrading their rating given this new information.

---

> > ### Author Response · Authors · 2024-11-24
> > **Follow-up**
> >
> > Dear Reviewer, as the discussion period approaches to an end, we would like to follow-up and check if our rebuttal adequately addressed your questions.

---

> > > ### Comment · Reviewer_brqJ · 2024-11-25
> > >
> > > Thank you very much for your clarification. After reading the rebuttal and other reviews, I raise my score to 6.

---

> > > > ### Author Response · Authors · 2024-11-25
> > > > **Thank you**
> > > >
> > > > Thank you for acknowledging our rebuttal and for raising your score.
> > > > Please let us know if there is anything we can do to make you consider increasing your rating further.
> > > > In any case, we sincerely thank you for helping us strengthen our submission.

---

### Official Review · Reviewer_MdgL · 2024-11-04

**Soundness:** 3
**Presentation:** 3
**Contribution:** 3
**Rating:** 6
**Confidence:** 3

**Summary:**

The paper investigates the impact of noisy data on (ambient) diffusion models. The paper shows that noisy data can be leveraged to improve the performance of diffusion models. In addition they provide the data regime where the noisy data are useful.

**Strengths:**

- the paper is well written
- the tackled problem is very interesting
- experiments are exhaustive and convincing (Figure 1 shows the regime of interest for using noisy data)

**Weaknesses:**

- I am not familiar with proof techniques for the mixture of Gaussian sample complexity, but I am not sure I understood the goal/take-away of Theorems 4.2-4.4. IMO much more discussion is needed here.
- Lack of motivation: the tackled problem is overall very interesting, but I am not sure that the authors provided a concrete application where such new noisy data are available, and known as such.

**Questions:**

- ¨For example, a blurry image might get dismissed from the filtering pipeline¨ Can authors provide a real example of such a filtering, i.e., a real filter used and a real discarded image?
- "For realistic sample sizes" Do you have an order of magnitude of "realistic sample size" in your context?
- Figure 1: in addition to training on noisy data, are other data augmentation techniques used? if not, do you know how new noisy data would compare against already existing standard data augmentation?


- The heteroscedastic model: I do not understand how the proposed model in Definition 4.1 is a heteroscedastic mixture of Gaussian. As currently written in the paper, it feels that all the points of the distribution are convolved with the Gaussian noise (which is coherent with the proposed setting). Following Shah (2023), shouldn't a heteroscedastic mixture of Gaussian write $X_i \sim \\mathcal{N}(\mu_i, \sigma_i)$.
In other words, IMO maths matches the setting, but I am not sure of the ¨heteroscedastic mixture of Gaussian¨ terminology.

- Regarding the cost of the augmentation, are all models trained with the augmentation, do you confirm that all models are trained with the same budget? (i.e., with the same number of images seen, since the datasets are different) I guess yes since EDM code takes ¨million number of images¨ as input


Theorem 4.2:
- what are $c_1$ and $c_2$? Absolute constants as defined in Vershynin (2018)?
- "Discussion" I did not understand the discussion after Theorem 4.2, could you elaborate on the "error in estimating the low-dimensional sub-space" and the "low-dimensional estimation procedures"
- "Then, there is a procedure which when given n independent ...." could you give a word on the procedure?

- Algorithm 1: is the noise level of each sample require in Algorithm 1?

---

> ### Author Response · Authors · 2024-11-20
> **Response to Reviewer MdgL**
>
> We thank the Reviewer for their time, feedback, and high-quality review. We are very glad that the Reviewer appreciated the strength of the submission. In what follows, we will address the remaining questions and concerns.
>
> ---
>
> **Feedback**: I am not sure I understood the goal/take-away of Theorems 4.2-4.4. IMO much more discussion is needed here.
>
>
> **Response**: That's a valid complaint and we updated our submission to address this. Theorems 4.2-4.4 study the usefulness of samples of differing quality in learning the parameters of a Gaussian mixture model. The Mixtures of Gaussians setting provides a concrete theoretical setting that can be studied to establish the trade-offs between high-quality and low-quality data. Our theoretical results establish tight bounds on the sample complexity which precisely characterize the utilities of high and low-quality samples. As stated in our paper, low-quality data is useful for learning high-level structural information about the distribution and high-quality data enables improved learning of local high-frequency details. These intuitions are made concrete through the results of Theorems 4.2-4.4. Following the Reviewer’s recommendation, we expanded the discussion of this section.
>
> Furthermore, these results motivate our notion of the effective sample size and its implications for data pricing in Section 5.4. These theorems thus provide a theoretical grounding for these definitions.
>
> ----
>
> **Feedback**: but I am not sure that the authors provided a concrete application where such new noisy data are available, and known as such.
>
> **Response**: We thank the Reviewer for the comment.
>
> The question of the Reviewer is where this “new noisy data” is coming from in real applications. Two concrete settings come to mind: i) synthetic data, and ii) data obtained from new experimental methods. Let’s take the domain of protein design as an example and discuss concretely these settings.
>
> **Synthetic Data**: Generative models for proteins are data-limited since there are only ~200K real proteins in the Protein Data Bank. The state-of-the-art models, such as Genie2 [1], are trained using synthetic data coming from proteins folded with Alphafold. The generated data from Alphafold have lower quality compared to real proteins. The authors of Genie use a hand-picked quality threshold (PLDDT > 80) for training only on Alphafold-generated data that have high prediction confidence. Because of this filtering, a lot of data is thrown away: particularly, from the 2.3M Alphafolded structures in AlphafoldDB, only ~600K of them are used for training of Genie.
>
> This is an example of an application where new noisy data become available (in this case from Alphafold). Instead of throwing away the data that do not pass the quality filters, our approach advocates for designing algorithms that use all of their data while accounting for the corruption of each datapoint.
>
> [1]: Genie 2: Designing and Scaffing Proteins at the Scale of the Structural Universe
>
> **New Experimental Methods**: As new experimental methods become available, it is easier to acquire data but sometimes the newly acquired data are of lower quality. For the example of the protein domain, traditionally protein structures that get deposited to the Protein Data Bank were acquired with X-ray crystallography. In the past few years, an alternative method, Cryo-Electron Microscopy has gained popularity. Cryo-EM structures often have a lower resolution, leading to imperfect structure determination [2]. The current state in Bio-ML models is to apply a handcrafted filtering pipeline (as in Alphafold) to filter proteins based on their resolution. Once again, our approach is that all samples have some information and we should not throw them away. Instead, the algorithms need to change to account for this newly available noisy data.
>
> [2]: Using deep-learning predictions reveals a large number of register errors in PDB depositions
>
> In the above, we provided to the Reviewer concrete applications where new noisy data is available. In both of these examples, there are a few really good data points (high-quality Alphafolded structures or proteins with high resolution), and there is plenty of noisy data as well.
> The noise model is far more complex than additive Gaussian noise and hence our techniques need to be generalized before becoming directly applicable to these settings. Closer to our setting is the MRI application, where the noise model can be well-approximated as a Gaussian corruption [3]. In any case, our paper shows that noisy data can be in principle quite valuable for learning, especially when a few clean data points are available.
>
> [3]: fastMRI: An Open Dataset and Benchmarks for Accelerated MRI
>
> ---
>
> (cont below)

---

> > ### Author Response · Authors · 2024-11-20
> > **Response (cont)**
> >
> > **Question**: Can authors provide a real example of such a filtering, i.e., a real filter used and a real discarded image?
> >
> >
> > **Response**:
> >
> > i) **Proteins**: For the filtering pipeline used in the state-of-the-art protein generative model, here are some examples of discarded proteins due to low PLDDT score:
> >
> > https://alphafold.ebi.ac.uk/entry/A0A023G4V4
> > https://alphafold.ebi.ac.uk/entry/A0A063Y3D8
> > https://alphafold.ebi.ac.uk/entry/A0A093VKP3
> >
> > As explained above, only 600K proteins are kept from the initial 2.3M pool.
> >
> > ii) **Images**: One of the most widely used datasets for images is Datacomp. Datacomp has an image-text alignment filtering pipeline. Below are some real images that got discarded from Datacomp due to bad image-text alignment:
> >
> >
> > https://ibb.co/y45tXnW
> >
> > The associated captions for these images are: ['NMC492323', 'O conjunto arquitet&ocirc;nico, projetado por Oscar Niemeyer em uma &aacute;rea de 84,5 mil metros quadrados, foi inaugurado em mar&ccedil;o de 1989.', 'Live @ the Forum', 'Clarette Щетка для распутывания волос DETANGLER Mini CDB 636 mtb road bike bicycle seatpost carbon fiber high strength seat tube cycling ultra light seatposts 27 2 30 8 31 6mm 350 400mm', 'I-COGROB_1']
> >
> > Further, we found many low-quality images in DataComp by asking CLIP to zero-shot classify between the "high-quality image" and "low-quality image" classes. Here are a few examples from Datacomp:
> >
> > https://ibb.co/PCtS01f
> >
> > Finally, we found that Datacomp used face-blurring to anonymize the dataset. Here are some found examples of blurred images in Datacomp:
> >
> > https://ibb.co/HPFBNVM
> >
> > ---
> >
> > **Question**: Do you have an order of magnitude of "realistic sample size" in your context?
> >
> > **Feedback**: We thank the Reviewer for the great question. In the paper, we tested dataset sizes up to ~2M images (ImageNet) and there is a significant quality degradation if you only train with noisy data as evidenced in Figure 1 and Tables 1, 2.
> >
> > ---
> >
> > **Question**:  Figure 1: in addition to training on noisy data, are other data augmentation techniques used? if not, do you know how new noisy data would compare against already existing standard data augmentation?
> >
> > **Response**: We thank the Reviewer for the question.
> >
> > All our experiments are based on the default settings from the EDM codebase that include augmentations for some datasets (e.g. CIFAR-10) and not for some others (e.g. CelebA). Whenever augmentations were used, they were used for all models, i.e. for the models trained with only clean, only noisy and a combination of clean and noisy.
> >
> > Further, we believe that this question is a unique opportunity for us to highlight a very important difference between augmentations and our noising process.
> >
> > In our noising process, each image is noised once (prior to the start of the training) and we get to see only this one particular noisy version of the image (i.e. the noise realization for a given image is fixed across training epochs).
> >
> > This is different compared to performing augmentations, because in the latter case each image in the dataset gets corrupted in different ways across different training epochs. Additionally to that, in typical augmentation pipelines, there is a considerable probability that the image will get corrupted to one epoch and not corrupted to the next one (i.e. augmentations do not throw away data). This is different compared to our case where the corruption happens once (before the training).
> >
> >
> > **Feedback**: The heteroscedastic model: I do not understand how the proposed model in Definition 4.1 is a heteroscedastic mixture of Gaussian.  In other words, IMO maths matches the setting, but I am not sure of the ¨heteroscedastic mixture of Gaussian¨ terminology.
> >
> > **Response**: This terminology has been adopted by prior work in a similar setting for the much simpler problem of mean estimation. The relationship between the two models follows from the observation that mean estimation corresponds to the Gaussian mixture models setting with a single component. We direct the reviewer to the following sources for further exploration of this topic.
> >
> > Luc Devroye, Silvio Lattanzi, Gábor Lugosi, Nikita Zhivotovskiy "On mean estimation for heteroscedastic random variables," Annales de l'Institut Henri Poincaré, Probabilités et Statistiques, Ann. Inst. H. Poincaré Probab. Statist. 59(1), 1-20, (February 2023)
> >
> > Compton, Spencer, and Gregory Valiant. "Near-Optimal Mean Estimation with Unknown, Heteroskedastic Variances." Proceedings of the 56th Annual ACM Symposium on Theory of Computing. 2024.
> >
> > ---
> >
> > (cont)

---

> > > ### Author Response · Authors · 2024-11-20
> > > **Response (cont)**
> > >
> > > **Question**: do you confirm that all models are trained with the same budget?
> > >
> > > **Response**: We confirm that all of the models were trained with the same budget, i.e. the total training steps were the same and the batch size and all the other hyperparameters were fixed.
> > >
> > > In fact, if we trained for longer the noisy models, we would get even more benefit (see Ablation for ImageNet), but we kept the training budget the same as in the EDM codebase for fair comparisons.
> > >
> > > ---
> > >
> > > **Question**: what are the constants?
> > >
> > > **Response**: Our constants are different from those mentioned in Vershynin (2018) and they are not explicitly computed in the proof.
> > >
> > > ---
> > >
> > > **Feedback**: could you elaborate on the "error in estimating the low-dimensional sub-space" and the "low-dimensional estimation procedures"
> > >
> > > **Response**: We apologize for the confusion. The procedure is a two-step procedure that first learns the low-dimensional subspace spanned by the means of the Gaussian mixture and then runs a different procedure (Algorithm 7) for estimating the means within the low-dimensional subspace. The errors in estimating the subspace are referred to as the “error in estimating the low-dimensional sub-space” and the algorithm employed after dimensionality reduction is the “low-dimensional estimation procedure”. We included additional details in the Discussion clarifying these error terms.
> > >
> > > ---
> > >
> > > **Question**: could you give a word on the procedure?
> > >
> > > **Response**: We apologize for the brevity due to space limitations. The procedure is specified in Algorithm 8 in Appendix C.3. As previously mentioned, the procedure is a two-step procedure that first learns the low-dimensional subspace and runs a different estimation procedure within the low-dimensional subspace once its been estimated. Algorithm 8 contains the details of the subspace estimation step while Algorithm 7 describes the low-dimensional estimation step. We included additional details in the Discussion clarifying the procedure yielding these results.
> > >
> > > ---
> > >
> > > **Question**: Algorithm 1: is the noise level of each sample require in Algorithm 1?
> > >
> > >
> > > **Response**: That’s a great question. We actually do not need to know the actual value of std since this can be estimated. Specifically, one can compute the difference of the variance between the clean and bad sets and this will give an estimation of the noise level of the noisy set. We added this discussion in the revised manuscript and we thank the Reviewer for helping us improve our work.

---

> ### Author Response · Authors · 2024-11-24
> **Follow-up**
>
> Dear Reviewer,
> as the discussion period approaches to an end, we would like to follow-up and check if our rebuttal adequately addressed your questions. Thank you for your time and input.

---

> > ### Author Response · Authors · 2024-11-28
> >
> > Dear Reviewer,
> > today is the last day to update the submission. We sincerely tried to answer all of your questions during the rebuttal in detail.
> > It would mean a lot to us if you could let us know if you find our answers satisfactory and if you would consider further increasing your positive rating.
> >
> > We remain at your availability throughout the discussion period to answer any remaining questions. Thank you for your time and feedback.

---

### Meta-Review · Area_Chair_xsgQ · 2024-12-17

**Metareview:**

The paper explores the challenge of training generative models when access to large-scale, high-quality datasets is limited. The authors demonstrate that a small subset of clean data (e.g., 10% of the total dataset) combined with a larger set of highly noisy data is sufficient to achieve performance comparable to models trained solely on similar-sized clean datasets, reaching near state-of-the-art results. To support these findings, the paper provides theoretical evidence through novel sample complexity bounds for learning from Gaussian Mixtures with heterogeneous variances. The theoretical analysis reveals that, for sufficiently large datasets, the effective marginal utility of a noisy sample is exponentially lower than that of a clean sample. This insight is further corroborated by experiments showing that a small clean subset significantly reduces the sample size requirements for noisy data.


Pros:
+ Important Problem Setting: The paper addresses the practical challenge of training diffusion models with noisy data, which is common in real-world applications.
+ Combination of Theory and Empirical Results: The work provides theoretical insights into sample complexity for Gaussian Mixture Models (GMMs) with heterogeneous variances, alongside extensive experiments demonstrating the effectiveness of using a small clean subset with a large noisy dataset.
+ Clear Presentation: The paper is well-written and logically structured, making the arguments and findings easy to follow.
+ Practical Utility: The findings offer significant implications for dataset curation strategies, balancing the use of noisy and clean data to achieve near state-of-the-art results.
+ Thorough Experiments: The study involves training over 80 models across datasets of varying noise levels, providing empirical validation of the claims.

Cons:
- Limited Novelty: The proposed algorithm and proof techniques are largely based on existing works, which raises questions about the paper's novelty.
- Narrow Scope of Analysis: The theoretical analysis is restricted to Gaussian noise and discrete distributions with finite support, limiting the applicability to more general noise models or continuous distributions.
- Gap Between Theory and Practice: The theoretical results focus on the Wasserstein loss, which does not perfectly align with the actual loss used in diffusion models, creating a disconnect between theory and implementation.
- Assumption of Known Noise Levels: The proposed method assumes that the noise level is known, which may not hold in practical scenarios. Addressing unknown noise levels would improve the method’s applicability.
- Simplistic Solution: While effective, the proposed solution is considered by some reviewers as overly simple and primarily engineering-driven.

After the revision as discussed below all reviewers recommend acceptance. I concur.

**Additional Comments On Reviewer Discussion:**

The discussion primarily revolved around the following points:

- Theoretical Contributions: Some reviewers raised concerns about the clarity and novelty of the theoretical results (Theorems 4.2–4.4). The authors clarified their significance and addressed ambiguities but acknowledged that they rely on existing techniques.

- Real-World Applications: Reviewers asked for concrete examples of where noisy data might arise and how it could be leveraged. The authors provided examples from protein design and experimental imaging methods, strengthening the paper's practical motivation.

- Experimental Comparisons: A few reviewers highlighted the lack of comparisons with previous works (e.g., Daras et al.). The authors argued that their method generalizes earlier work by introducing clean data into the mix, demonstrating superior performance.

- Noise Level Assumptions: The assumption of known noise levels was questioned. The authors responded that noise levels can be estimated, but reviewers suggested incorporating this into the theoretical analysis.

The discussion led to a slight convergence of opinions, with some reviewers increasing their scores and all reviewers recommending acceptance.

---

### Decision · Program_Chairs · 2025-01-22

Accept (Poster)